EMBO

reports

# Report

# WIPI2b recruitment to phagophores and ATG16L1 binding are regulated by ULK1 phosphorylation

Andrea Gubas [ID] [1,8,9], Eleanor Attridge [ID] [1,9], Harold BJ Jefferies[1], Taki Nishimura[2,3], Minoo Razi[1], Simone Kunzelmann [ID] [4], Yuval Gilad[5], Thomas J Mercer[6], Michael M Wilson[7], Adi Kimchi [ID] [5] & Sharon A Tooze [ID] [1✉]

## Abstract

**One of the key events in autophagy is the formation of a double-membrane phagophore, and many regulatory mechanisms underpinning this remain under investigation. WIPI2b is among the first proteins to be recruited to the phagophore and is essential for stimulating autophagy flux by recruiting the ATG12–ATG5–ATG16L1 complex, driving LC3 and GABARAP lipidation. Here, we set out to investigate how WIPI2b function is regulated by phosphorylation. We studied two phosphorylation sites on WIPI2b, S68 and S284. Phosphorylation at these sites plays distinct roles, regulating WIPI2b's association with ATG16L1 and the phagophore, respectively. We confirm WIPI2b is a novel ULK1 substrate, validated by the detection of endogenous phosphorylation at S284. Notably, S284 is situated within an 18-amino acid stretch, which, when in contact with liposomes, forms an amphipathic helix. Phosphorylation at S284 disrupts the formation of the amphipathic helix, hindering the association of WIPI2b with membranes and autophagosome formation. Understanding these intricacies in the regulatory mechanisms governing WIPI2b's association with its interacting partners and membranes, holds the potential to shed light on these complex processes, integral to phagophore biogenesis.**

**Keywords** Autophagy; Autophagosome; Kinase; WIPIs; Amphipathic Helix
**Subject Categories** Autophagy & Cell Death; Signal Transduction

## Introduction

To maintain homoeostasis or survival during stress, eukaryotic cells rely on autophagy. Autophagy is a conserved catabolic process, in which portions of cytoplasm, damaged organelles, pathogens or protein aggregates are delivered to the lysosome via double-membraned vesicles termed autophagosomes. Autophagosomes are formed de novo at the ER (Nishimura and Tooze, 2020). A key event in autophagosome biogenesis is the formation of a double-membrane phagophore, the mechanism of which remains incompletely understood. The phagophore elongates, sequestering cellular components marked for degradation, and eventually closes, forming an autophagosome. Autophagosomes fuse with endocytic organelles, including late endosomes, and, finally with the lysosome, in which the sequestered cargo is degraded and recycled back into the cytosol.

Autophagy is upregulated upon amino acid starvation. When the essential growth control regulator, mTOR (mammalian Target of Rapamycin), is inactivated, the Unc-51-like kinase (ULK) complex is activated, leading to subsequent ULK1 and ULK2 autophosphorylation and phosphorylation of its subunits ATG13, FIP200, and ATG101 (Mercer et al, 2018), and translocation from the cytosol to membranes where autophagosome formation is initiated (Chan et al, 2009; Ganley et al, 2009; Hara et al, 2008; Karanasios et al, 2013). Activated ULK1/2 phosphorylates ATG9A at Serine 14, driving translocation of ATG9A to the phagophore initiation site ((Zhou et al, 2017)). ULK1 also phosphorylates all members of the class III Phosphatidylinositol (3) kinase (PI3K) complex I, Beclin1, ATG14L, VPS34, and VPS15 (Mercer et al, 2021; Ohashi, 2021), leading to activation of lipid kinase activity. VPS34 generates pools of Phosphatidylinositol-3-phosphate (PI3P) at the initiation site, driving the formation of omegasomes, which serve as cradles for phagophore formation (Axe et al, 2008; Mercer et al, 2018). The ULK complex is retained on the omegasomes through direct association of the C-terminal domain of ULK1/2 with membranes (Chan et al, 2007), and with ATG13 (Hieke et al, 2015), which has been shown to bind PI3P (Karanasios et al, 2013). Finally, ULK1 phosphorylates a key regulator of autophagosome maturation, Syntaxin 17 (Wang et al, 2018). ULK1 and ULK2 are functionally redundant (McAlpine et al, 2013) and while they may have overlapping substrates, there is evidence they have distinct activities, including substrates, in different tissues and cells (Demeter et al, 2020).

WIPI proteins, a family of 4 proteins WIPI1-4, are also recruited by PI3P and play a fundamental role in phagophore maturation (Dooley et al, 2014; Fracchiolla et al, 2020; Gaugel et al, 2012; Polson et al, 2010). WIPI2b, a WIPI2 isoform, binds ATG16L1, together with the ATG12~ATG5 ubiquitin-like conjugate, acts as an E3 ligase directing lipidation of the ATG8 family members LC3 and

[1]Molecular Cell Biology of Autophagy, The Francis Crick Institute, 1 Midland Road, London NW1 1AT, UK. [2]Department of Biochemistry and Molecular Biology, Graduate School and Faculty of Medicine, The University of Tokyo, Tokyo, Japan. [3]PRESTO, Japan Science and Technology Agency, Chiyoda-ku, Tokyo, Japan. [4]Structural Biology Science Technology Platform, The Francis Crick Institute, 1 Midland Road, London NW1 1AT, UK. [5]The Weizmann Institute of Science, Rehovot, Israel. [6]Genetech, 1 DNA Way, South San Francisco, CA 94080, USA. [7]The Babraham Institute, Cambridge CB22 3AT, UK. [8]Present address: Muscular Dystrophy UK, London SE1 8QD, UK. [9]These authors contributed equally: Andrea Gubas, Eleanor Attridge. ✉E-mail: Sharon.tooze@crick.ac.uk

GABARAP on the forming phagophore (Dooley et al, 2014; Fracchiolla et al, 2020; Lystad et al, 2019). This is further facilitated by the autophagy receptor Optineurin, through its binding to WIPI2 and the ATG12–ATG5–ATG16L1 complex (Bansal et al, 2018).

WIPI2 is a member of the PROPPIN (β-propellers that bind phosphoinositides) family (Polson et al, 2010; Proikas-Cezanne et al, 2004). PROPPINs fold as seven-bladed β-propellers and contain two PI3P-binding sites found in the conserved F/LRRG motif in blades 5 and 6 of the propeller (Baskaran et al, 2012; Jensen et al, 2022; Krick et al, 2012; Watanabe et al, 2012). In all WIPI proteins there is a hydrophobic loop in blade 6 which is predicted to form an amphipathic helix (Gopaldass et al, 2017). These features are thought to be required for membrane association of the PROPPINs. However, despite inhibition of PI3P production, and mutation of the FRRG motif, a small pool of WIPI2 remains membrane associated under fed conditions (Polson et al, 2010).

WIPI2 is subject to alternative splicing, resulting in six different isoforms or splice variants—WIPI2a, b, c, d, e and f. The functions of these isoforms differ in autophagy, and thus far only WIPI2b and WIPI2d have been found to be recruited to phagophores, interact with ATG16L1 and drive autophagic flux (Dooley et al, 2014; Strong et al, 2021). WIPI2b and WIPI2d are identical apart from an 11 amino acid stretch within the WIPI2b flexible C-terminus, which is missing in WIPI2d (Proikas-Cezanne and Robenek, 2011). While WIPI2 orchestrates the early phases of autophagosome biogenesis via direct binding of key autophagy protein complexes and signalling lipids, the precise regulation of these functions remains elusive. Upon amino acid starvation, and production of PI3P, WIPI2 forms punctate structures in cells which contain early autophagic markers (Dooley et al, 2014; Karanasios et al, 2013; Polson et al, 2010; Proikas-Cezanne and Robenek, 2011). WIPI2b co-localises with ULK1 puncta (Kraft et al, 2012), and co-immunoprecipitates with ULK1 (Gilad and Kimchi, 2014), suggesting that ULK1 may be directly involved in regulation of WIPI2b function during autophagy. Wan and colleagues reported that WIPI2b is phosphorylated by mTOR at S395, which drives WIPI2b binding to the E3 ligase HUWE1, as well ubiquitination and proteasomal degradation (Wan et al, 2018). WIPI2b has also been shown to be ubiquitinated and targeted for degradation during mitosis, through CRL4 (Lu et al, 2019). These results support the hypothesis that WIPI2b function may be regulated by phosphorylation and ubiquitination.

We show that WIPI2b is a novel substrate of the ULK1 kinase. We identified six phosphorylation sites, of which S68 and S284 are critical sites for regulating WIPI2b interaction with ATG16L1 and membrane association, respectively. These data provide mechanistic insights into early autophagosome biogenesis and link two key events associated with the role of WIPI2b in autophagy: its interaction with ATG16L1 and association with phagophores.

# Results and discussion

## WIPI2b interacts with ULK1

Understanding the processes underlying phagophore biogenesis remains an important question. Recruitment of WIPI2b to PI3P on the omegasome, and its interaction with ATG16L1, are essential for phagophore nucleation and subsequent autophagosome formation

(Dooley et al, 2014; Strong et al, 2021). Regulation of WIPIs activity by, for example, phosphorylation would be critical to ensure timely autophagic flux.

To investigate whether ULK1 interacts with WIPI2, we immunopurified endogenous WIPI2 from HEK293A cells. ULK1 was detectable in the immunoprecipitate (Fig. 1A), consistent with the previous report using tagged proteins (Gilad and Kimchi, 2014). The WIPI2 antibody does not discern between the six WIPI2 splice isoforms (WIPI2a-f) (Proikas-Cezanne et al, 2015), however, we focused on WIPI2b due to its localisation to phagophores and its known association with ATG16L1 (Dooley et al, 2014; Lystad et al, 2019; Strong et al, 2021). ATG16L1 also interacts with FIP200, a subunit of ULK1 complex (Dooley et al, 2014; Fujita et al, 2013; Gammoh et al, 2013; Lystad et al, 2019; Nishimura et al, 2013). To test if the interaction between ULK1 and WIPI2b is mediated through ATG16L1 binding to FIP200, we analysed ULK1 binding to WIPI2b when ATG16L1 binding was abolished by mutating residues WIPI2 Arg108 and Arg125 to glutamic acid residues (RERE mutant) (Dooley et al, 2014). Co-immunoprecipitation was performed in HEK293A WIPI2 KO cells (Fig. EV1) and rescued by transfection with GFP-WIPI2b WT or GFP-WIPI2b RERE (Fig. 1B). ULK1 and ATG13 bound GFP-WIPI2b WT (wild-type), and this interaction was increased when the binding of WIPI2b to ATG16L1 was abolished (Fig. 1B,C).

ULK1, ATG13 and ATG16L1 binding to GFP-WIPI2b WT occurred independently of nutrient starvation (Fig. 1D). However, mutation of the FRRG motif to FTTG in WIPI2b, which abolishes binding to PI3P, decreased the binding of ULK1 and ATG13, suggesting that the interaction between WIPI2 and the ULK complex might be more stable at the phagophore (Fig. 1D,E). In contrast, ATG16L1 binding was unaffected, suggesting ATG16L1 binding to WIPI2b occurs independently of phagophore membrane association.

We also tested if the binding of ULK1 to WIPI2b was dependent on kinase activity of ULK1. We pulled down GFP-WIPI2b from cells transfected with either WT or kinase-inactive mutant (K46I, KI) ULK1. Both WT and ULK1 KI interacted with GFP-WIPI2b, although the ULK1 KI showed a slight, but significant decrease in binding to GFP-WIPI2b (Fig. 1F,G). This suggests that the kinase activity may help stabilise the interaction between ULK1 and WIPI2b.

## Identification of novel ULK1-dependent phosphorylation sites on WIPI2b

As WIPI2b interacts with ULK1, and the binding could be affected by the ULK1 activity (Fig. 1F,G), we hypothesised that WIPI2b may be an ULK1 substrate. WIPI2b interaction with ATG16L1 and its association with PI3P and membranes is mediated by electrostatic interactions (Dooley et al, 2014; Strong et al, 2021), which can likely be disrupted by post-translational modifications, such as phosphorylation. According to PhosphositePlus (https://www.phosphosite.org/), WIPI2 is reported to be phosphorylated at several sites (Hornbeck et al, 2015).

We explored WIPI2b phosphorylation by observing shifts in migration patterns by SDS-PAGE and western blot. Phosphorylated proteins have a greater negative charge and can migrate slower on SDS-PAGE gel than non-phosphorylated proteins. By Western blot, both endogenous WIPI2 (Fig. 2A) and overexpressed HA-WIPI2b

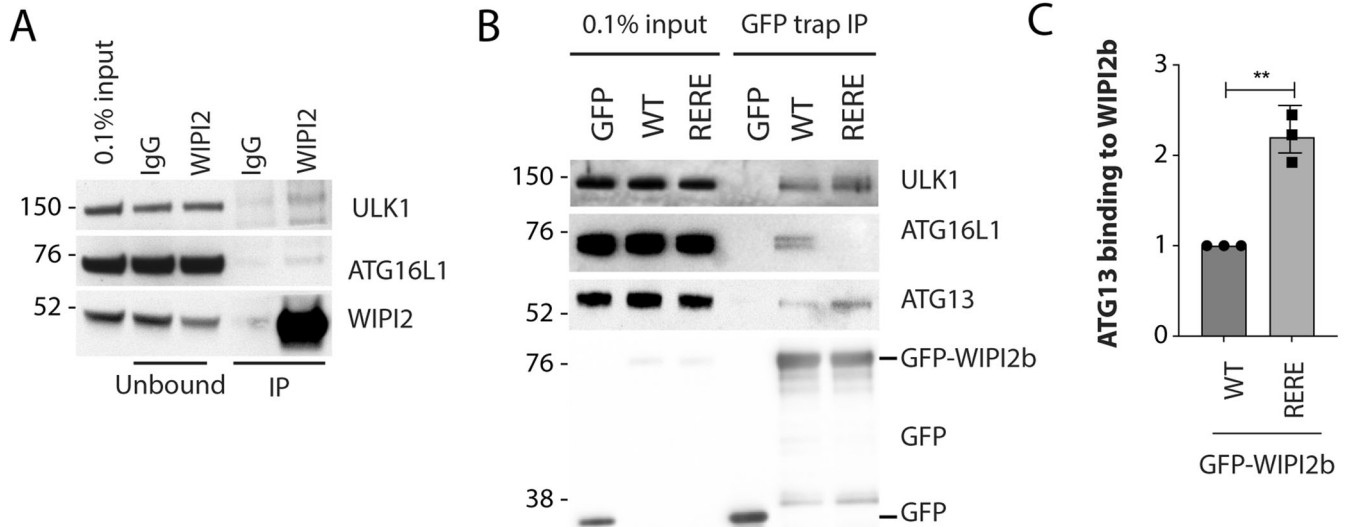

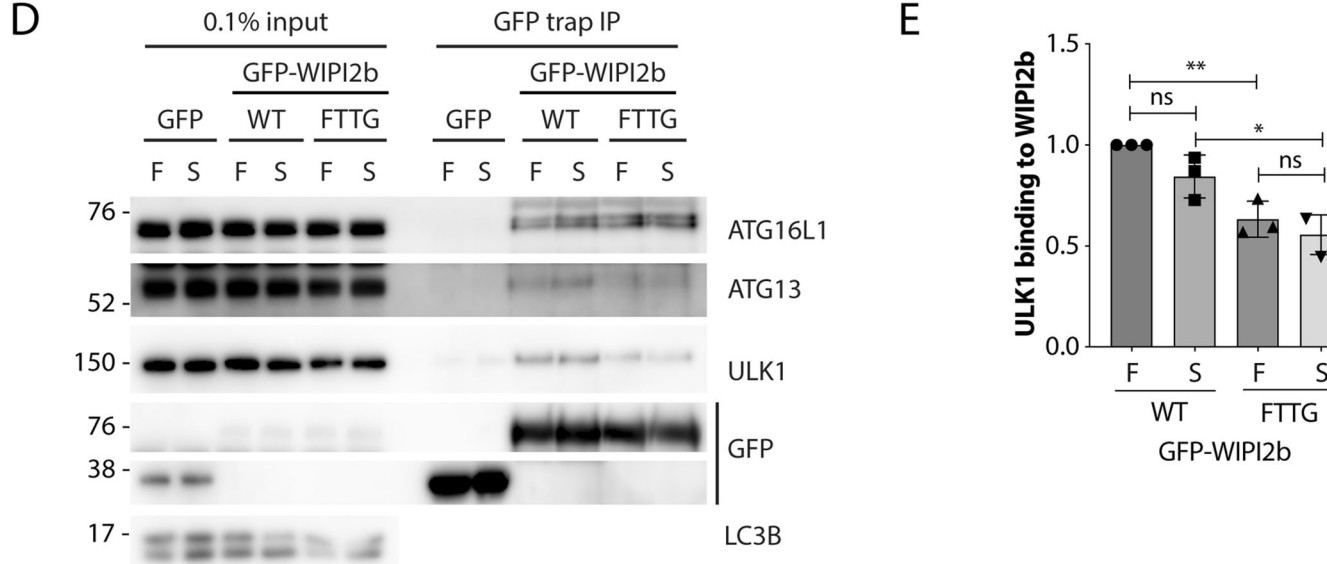

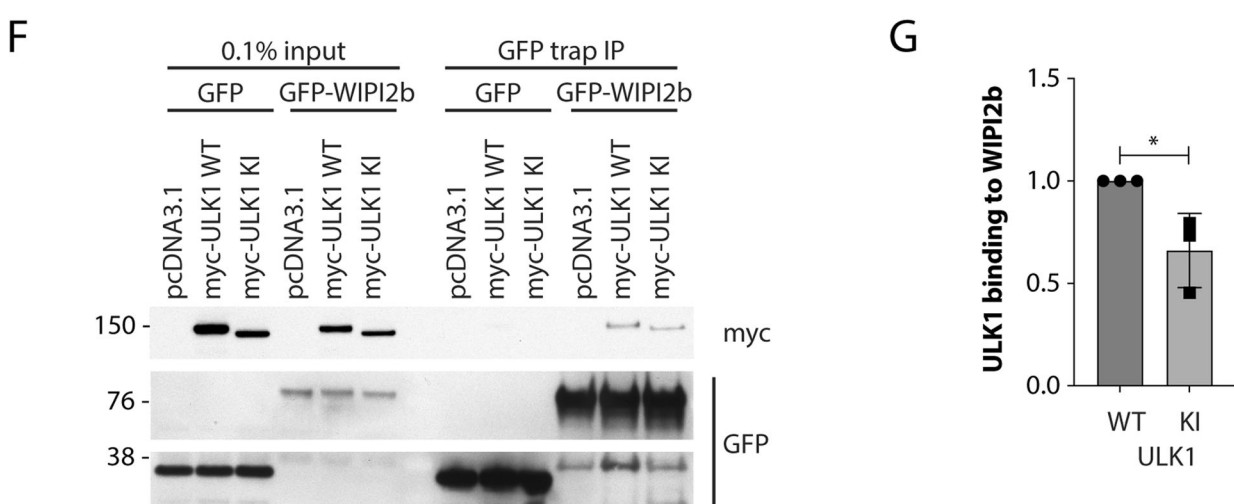

Figure 1. WIPI2b interacts with ULK1.

(A) Endogenous WIPI2 was immunoprecipitated using WIPI2 monoclonal antibody crosslinked to protein G (Pt G) sepharose beads using DMP (Dimethylpimelimidate) crosslinker. Mouse IgG crosslinked to Pt G sepharose was used as a negative control. One repeat of $n = 3$ is shown. (B) GFP, GFP-WIPI2b WT or RERE mutants were expressed and pulled down from HEK293 cells, and western blot analysis for ULK1, ATG13 and ATG16L1. One repeat of $n = 3$ is shown. (C) Quantification of (B) mean ± SEM from $n = 3$ biological replicates, unpaired Student's $t$ test, **$P = 0.0014$. (D) GFP, GFP-WIPI2b WT or FTTG mutant were expressed, and 24 h later the cells were incubated in full medium (F) or starved in EBSS for 2 h (S). Western blot analysis was done with indicated antibodies. (E) Quantification of (D) mean ± SEM from $n = 3$ biological replicates, one-way ANOVA with Tukey's post test, **$P = 0.0032$, *$P = 0.0135$. (F) GFP-Trap of GFP or GFP-WIPI2b WT co-expressed with myc-ULK1 WT or kinase-inactive (KI) myc-ULK1. Note, the KI typically expresses at a lower level and the data was normalised to WIPI2. (G) Quantification of (F), mean ± SEM from $n = 3$ biological replicates, unpaired Student's $t$ test, *$P = 0.037$. Source data are available online for this figure.

WT (Fig. 2B) migrated as a doublet which collapsed into a single lower molecular weight band after lambda phosphatase treatment, suggesting that a pool of WIPI2b is constitutively phosphorylated. EBSS (Earle's buffered saline solution) treatment to induce amino acid starvation had no effect on the overall migration pattern of endogenous WIPI2 (Fig. 2A). To identify phosphorylation sites on WIPI2b that are dependent on ULK1 activity, we transiently expressed HA-ULK1 WT or HA-ULK1 KI with or without GFP-WIPI2b WT, followed by immunoprecipitation (IP) of endogenous WIPI2 or GFP-tagged WIPI2b (Fig. EV2A,B). Mass spectrometry analysis of the IPs revealed a number of phosphorylation sites on GFP-WIPI2b WT that were dependent on co-expression with active ULK1, including S39, S68, S96, S185, S284 and S360, as well as S395, of which S68, S96, S360 and S395 were also detected on endogenous WIPI2 (Fig. 2C). In vitro Kinase assays (immunopurified myc-ULK1 and purified WIPI2b, see Fig. 5B–D) analysed by mass spectroscopy confirmed phosphorylation of S68 and S284 (Dataset EV1).

To confirm ULK1-dependent phosphorylation of WIPI2, HA-tagged WIPI2b was co-expressed with either myc-tagged ULK1 WT or KI in HEK293A cells, followed by 2 h amino acid starvation in EBSS to further activate ULK1 (Fig. 2D). Mouse myc-ULK1 WT overexpression decreased the mobility of WIPI2b-HA, which is indicative of protein phosphorylation, while myc-ULK1 KI appeared to have no overall effect. Note that the KI is typically expressed at lower levels. These results were also replicated with human HA-ULK1 WT and KI co-expressed with myc-WIPI2b WT (Fig. EV2C). To further support the phosphorylation of WIPI2 by ULK1, we treated cells with inhibitors of ULK1, SBI-0206965 (Egan et al, 2015) and MRT68921 (Petherick et al, 2015). HEK293A cells transiently co-expressing HA-WIPI2b WT, and myc-ULK1 WT, were treated for 1 h with EBSS, or EBSS with SBI-0206965 or MRT68921, or EBSS followed by lambda phosphatase treatment post-lysis (Fig. 2E). We observed a similar change in mobility with both inhibitors as seen with ULK1 KI. Together these data suggest that WIPI2b is phosphorylated by ULK1 at a number of sites.

## Characterisation of WIPI2b phosphorylation sites

The phosphorylation motifs of Atg1 and ULK1 are characterised by hydrophobic residues at positions −3, +1 and +2 (Egan et al, 2015; Papinski and Kraft, 2016). Alignment of six WIPI2b phosphorylation sites identified upon ULK1 overexpression revealed that the six sites roughly conform to the ULK1 phosphorylation motif, with S68 being closest to the optimal motif (Fig. 2F). We mutated all six to alanine (6A) or aspartate (6D) to mimic fully non-phosphorylated or phosphorylated proteoforms, respectively. We co-expressed the WIPI2b WT or mutants either with ULK1 WT or ULK1 KI and

assayed their migration shift (Fig. 2G). Co-expression of WIPI2b-HA 6A and ULK1 WT led to a minor change in the mobility of 6A compared to WIPI2b WT, suggesting that the six mutated sites could be ULK1 sites. When WIPI2 6D is expressed alone or with ULK1, it displayed a mobility shift comparable to the WT co-expressed with ULK1 WT supporting the notion that the slowest migrating band is phosphorylated at multiple sites. To further support that WIPI2b is phosphorylated by ULK1 at the six phosphorylation sites, we performed an in vitro kinase assay, using WIPI2b WT and WIPI2b 6A (Fig. 2H). These results further corroborate the conclusion that WIPI2b is phosphorylated by ULK1, and likely at 6 different sites.

S395 is thought to be an mTORC1-dependent phosphorylation site on WIPI2b and is consistently identified in mass spectrometry experiments with high confidence (TYVPSS$_{395}$PTRL) (Hsu et al, 2011; Wan et al, 2018). As such, it was excluded from our chosen set of phosphorylation sites. To investigate if the small mobility changes of WIPI2b WT or 6A (see Fig. 2G) is due to S395 phosphorylation, S395 was mutated in WIPI2b 6A, therefore generating WIPI2b 7A mutant (Fig. EV2D). WIPI2b 7A mutant runs as a single band on the gel, even upon ULK1 WT overexpression. A comparison of the ULK1-dependent mobility shift of WIPI2b 6A and WIPI2b 7A in Fig. EV2D suggests the slowest migrating band in the WIPI2b 6A corresponds to phosphorylated S395. Since the S395 mutation to alanine in WIPI2b 6A completely abolished the faint slower migrating band, we confirmed that this molecular weight shift is due to S395 phosphorylation by mutation of S395 to an alanine (Fig. EV2E).

## Phosphomutants of WIPI2b cannot rescue autophagy in WIPI2 KO cells

As ULK1 phosphorylates WIPI2b in vitro and in vivo, we investigated the role of ULK1 phosphorylation on WIPI2b function in autophagy. Using the WIPI2 KO HEK293A cell line, we expressed empty vector (EV), WIPI2b WT, 6A and 6D mutants and measured autophagic flux using LC3 lipidation in starvation in the presence or absence of Bafilomycin A1 (Baf A1). Surprisingly, as we believed ULK1 phosphorylation would be activating, LC3B lipidation was decreased in phospho-mimic WIPI2b 6D compared to WIPI2b WT and WIPI2b 6A (Fig. 3A,B). A hallmark of autophagy is the formation of WIPI puncta recruited to PI3P-rich phagophores via the FRRG motif (Jeffries et al, 2004; Proikas-Cezanne et al, 2004). As WIPI2b also responds to starvation by forming puncta (Polson et al, 2010), we tested if this was affected by mutation of the six putative ULK phospho-sites (Fig. 3C,D). The WIPI2b 6A mutant formed more puncta in fed conditions than WT, and this was further increased upon amino acid starvation.

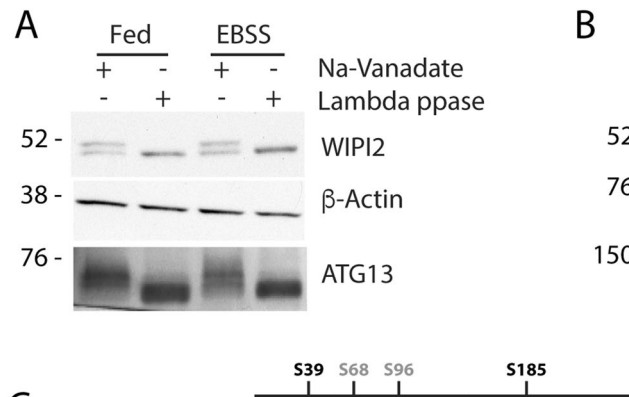

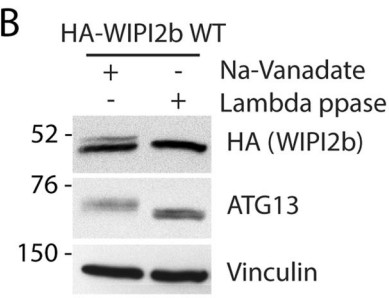

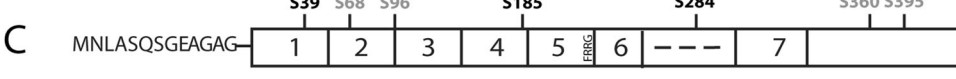

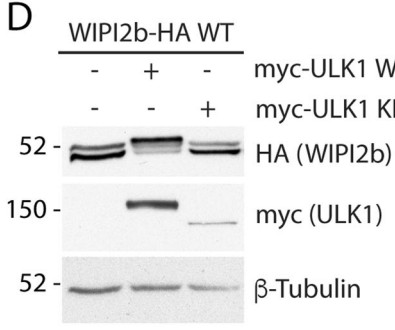

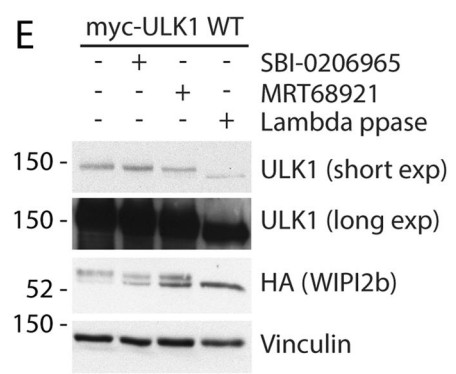

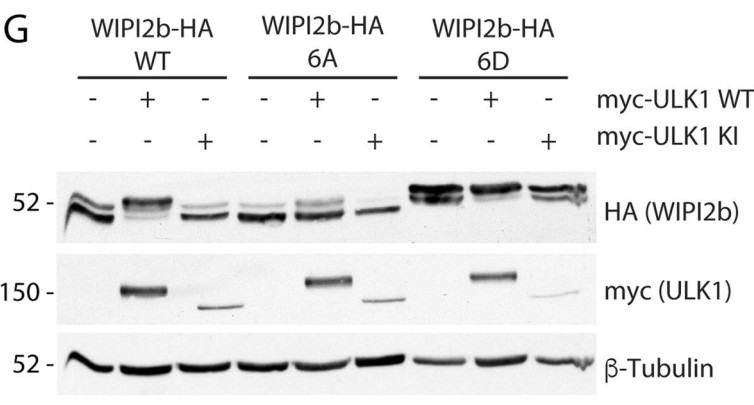

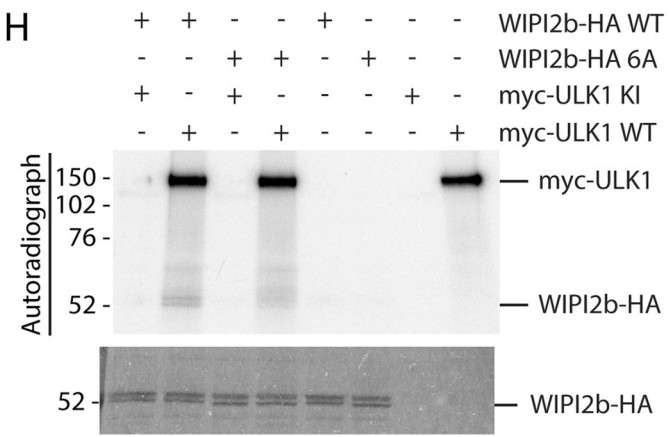

◀  **Figure 2.  ULK1 phosphorylates WIPI2.**

(A, B) Untransfected (A) or HA-WIPI2b transfected (B) HEK293A cells, were in full medium (B), or in full medium (fed) or starved (EBSS) for 2 h (A). Cell lysates were incubated with Na-Vanadate or Lambda phosphatase for 30 min, western blotted with antibodies against WIPI2, ATG13, HA or loading controls (β-actin and vinculin). Representative experiment from $n = 2$ (A) and $n = 3$ (B). (C) Features of WIPI2b: 1–7 WD40 repeats; dashed line the unstructured region; C-terminal unstructured region. Phosphorylation sites identified on GFP-WIPI2b (black and grey) or endogenous WIPI2 (grey only). (D) WIPI2b-HA expressed with myc-ULK1 WT or KI, starved for 1 h in EBSS and subjected to western blot with indicated antibodies. Representative experiment from $n = 3$. (E) HA-WIPI2b expressed with myc-ULK1 WT, incubated with SBI-0206965, MRT68921, or lambda phosphatase. Western blot performed with indicated antibodies. Representative experiment from $n = 3$. (F) Alignment of the six WIPI2b phospho-serines. The ULK1 consensus motif is displayed above with hydrophobic residues underlined. (G) Cells co-expressing WIPI2b-HA WT, WIPI2b-HA 6A or WIPI2b-HA 6D and myc-ULK1 WT or myc-ULK1 KI, treated in EBSS for 2 h, analysed by western blot with indicated antibodies. (H) HEK293A cell lysates expressing WIPI2b-HA WT, WIPI2b-HA 6A, myc-ULK1 WT or myc-ULK1 KI were immunoprecipitated with HA or myc antibodies bound to beads. The beads were mixed as indicated for kinase assay with ($^{32}$P)ATP, followed by SDS-PAGE and autoradiography. Representative experiment of $n = 3$. Source data are available online for this figure.

The WIPI2b 6D mutant was unable to form puncta in either fed or starved conditions, suggesting that phosphorylation could disrupt the association of WIPI2b with PI3P and/or membranes.

To understand if the lack of WIPI2b 6D puncta was a result of decreased association with membranes, WIPI2b-HA WT, WIPI2b-HA 6A and WIPI2b-HA 6D were expressed in HEK293A cells and a crude membrane fractionation was performed (Fig. EV2F,G). As previously shown a population of WIPI2 is present on membranes in both fed and starved conditions (Polson et al, 2010), and as expected WIPI2b WT is found on membranes, and we found the 6A membrane population was increased relative to WT.

Together, these data show that phosphorylation of WIPI2b at the identified six sites may negatively regulate WIPI2b's function in autophagy by preventing WIPI2 association to membranes, and puncta formation. These conclusions are based on multiple, six, phosphosite mutations and viewed with caution as the phospho-mutations will have introduced significant charge changes.

To understand which site(s) may be responsible for the effect on autophagic flux, and to overcome the limitations of multiple mutations, we focused on two sites which we predicted might impact the known properties of WIPI2b: S68, which could impair the binding of the E3 ligase ATG12~ATG5-ATG16L1 (Dooley et al, 2014), and S284, which is present in a predicted amphipathic α-helix (AH) (Gopaldass et al, 2017) and may impair recruitment to PI3P-rich domains.

## Phosphorylation at WIPI2b S68 regulates its interaction with ATG16L1

We predicted that S68, which is in the proximity of the hydrophobic region where ATG16L1 binds to WIPI2b, based on homology with Hsv2 (Dooley et al, 2014), may have a regulatory role. In the structure of WIPI2d, S68 is found within an exposed loop between beta sheets 1 and 2 in blade 2, which confirms this region is required for the interaction with ATG16L1 (Strong et al, 2021). Strong et al also found that S68 contributes towards stabilising the binding with ATG16L1. As the interaction between WIPI2b and ATG16L1 is mediated by electrostatic interactions, R108 and R125 on WIPI2b and E226 and E230 on ATG16L1 (Dooley et al, 2014), introduction of a negative charge on this binding surface through phosphorylation of WIPI2b at S68 likely displaces ATG16L1, rendering WIPI2b unable to drive lipidation of LC3 or GABARAP proteins. Our modelling (Fig. EV3A) suggests that a phosphate residue on S68 protrudes well above the surface of the WIPI2d ATG16L1-binding pocket, sterically hindering binding of the ATG16L1 helix backbone residues (227–229) of the WIPI2 interacting region. The interactions of residues proximal to pS68,

including H85 (red arrowhead) and K88 (blue arrowhead), may also be disrupted (Fig. EV3A). Alignment of WIPI1 and WIPI2 orthologs from different species shows that S68, and the surrounding region, is highly conserved from yeast to human (Fig. 4A). Co-expression of ULK1 with WIPI2b S68A revealed a minor band migrating faster compared to WIPI2b WT, supporting WIPI2b phosphorylation at S68 (Fig. 4B).

To determine if there is a functional impact of S68 phosphorylation of WIPI2b, we examined whether phosphorylation affects ATG16L1 binding to S68A and S68D mutants. Co-IP experiments showed that, compared to WIPI2b WT, binding of ATG16L1 to WIPI2b S68A is reduced by ~30%, while binding to S68D is reduced by more than 60% (Fig. 4C,D), suggesting that phosphorylation of WIPI2b at S68 regulates WIPI2b binding to ATG16L1.

In WIPI2 KO cells, ATG16L1 puncta are unable to form upon starvation (Fig. 4E), and rescue with WIPI2b WT and WIPI2b S68A in WIPI2 KO cells restored ATG16L1 puncta, while WIPI2b S68D failed to rescue ATG16L1 puncta (Fig. 4E,F). In line with the loss of ATG16L1 binding and absence of ATG16L1 puncta after rescue with WIPI2b S68D mutants, this mutant also failed to rescue LC3 puncta formation (Figs. 4G and EV3B). LC3 puncta data was supported by flux analysis: WIPI2b S68D mutant expression is unable to rescue LC3 lipidation in WIPI2 KO cells, compared to both WIPI2b WT and S68A (Fig. 4H,I).

WIPI2 puncta formation in WIPI2 KO cells was found with WIPI2b WT and S68A, but not with WIPI2b S68D (Fig. EV3C,D). Previous work mapping interactors with the yeast homologue Atg18 (Rieter et al, 2013), a PROPPIN which folds into a seven-bladed β-propeller, and the structural work on WIPI2d (Strong et al, 2021), suggests that the top of the propeller (blades 1–4) is not involved in interaction with membranes, but rather mediates protein-protein interactions. Phosphorylation of S68, positioned on the top of the propeller structure, disrupts ATG16L1 binding: the S68D mutation forms very few puncta relative to WT and S68A proteins (Fig. EV3C,D). This phenomenon was also observed by Strong et al in vitro, where two of the ATG16L1-binding mutants of WIPI2d (K88E and K128E) failed to be recruited to PI3P-enriched liposomes (Strong et al, 2021). It is not clear why these mutations in the ATG16L1-binding surface disrupt the interaction of WIPI2 with the membrane. We speculate that the decreased number of WIPI2b S68D puncta could be a consequence of reduced clustering of WIPI2b at the phagophore due to the lack of binding to ATG16L1, as previously suggested in yeast (Juris et al, 2015), thus destabilising WIPI2b on the membranes.

Phosphorylation at S68 may also have regulatory functions for other interactors of WIPI2b. R108, R125 and K88 in WIPI2d are predicted to form electrostatic interactions with the mitophagy

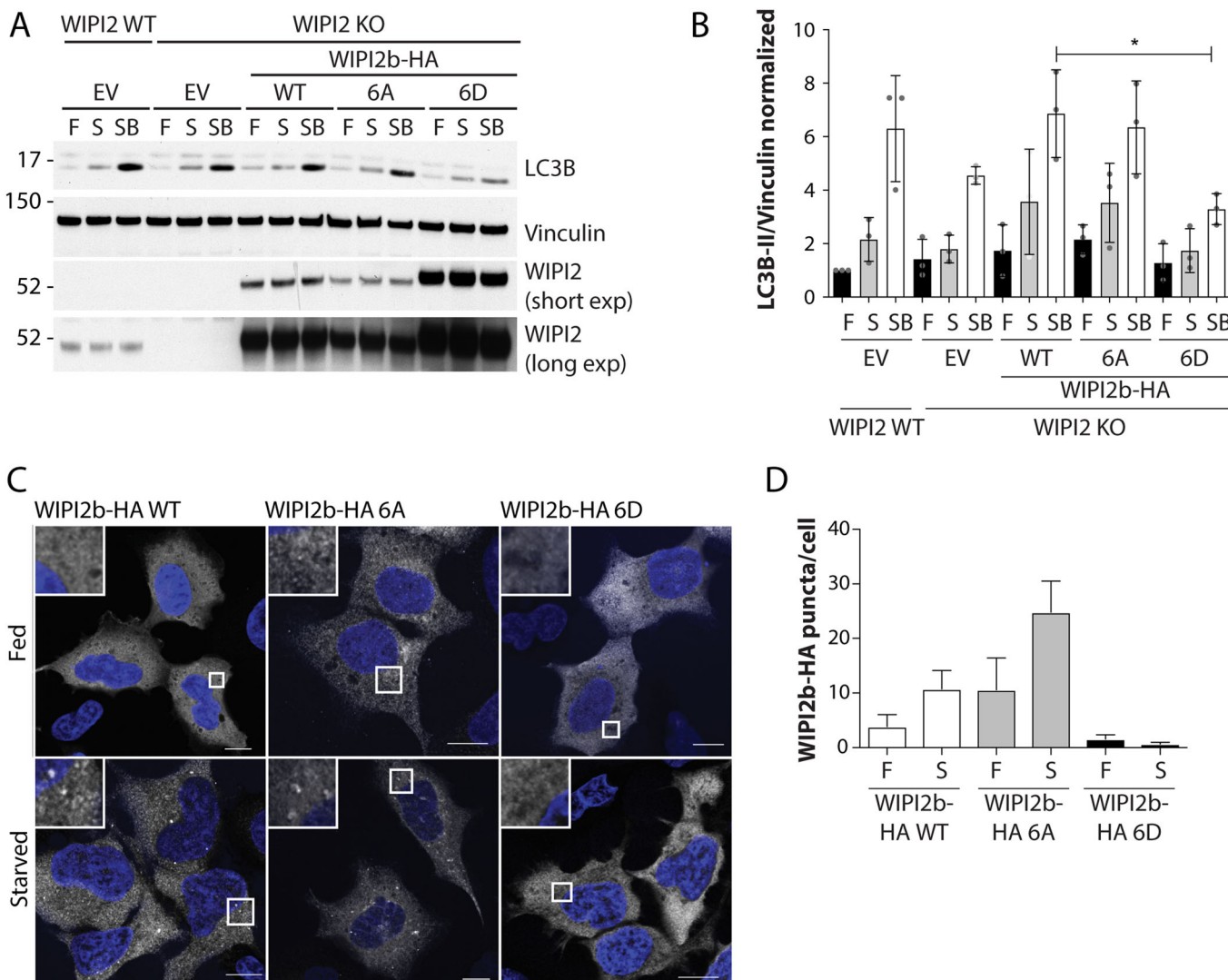

**Figure 3. Mutation of multiple phosphorylation sites in WIPI2b abolishes autophagy.**

(A) WIPI2 CRISPR WT cells expressing EV and WIPI2 CRISPR KO cells expressing EV, WIPI2b-HA WT, WIPI2b-HA 6A or WIPI2b-HA 6D were incubated 2 h in full medium (F), EBSS (S) or EBSS with Bafilomycin A1(SB) and analysed for LC3 lipidation by western blot. (B) Quantification of LC3-II/vinculin in (A) SEM for $n = 3$ biological replicates, one-way ANOVA with Tukey's post test, $*P = 0.041$. (C) WIPI2b-HA WT, WIPI2b-HA 6A and WIPI2b-HA 6D were expressed in HEK293A cells and treated for 2 h in EBSS (Starved) or left untreated (Fed) and labelled with HA antibodies. Scale bar = 10 μm. (D) Quantification of HA-WIPI2-positive puncta per cell from (C). SEM from ten frames per condition per experiment $n = 1$ biological repeat. Source data are available online for this figure.

protein Nix (Bunker et al, 2023). This interaction is required for WIPI2 recruitment to Nix in Nix/BNIP3 mitophagy. Phosphorylation at S68 in WIPI2b may also disrupt binding to Nix to regulate mitophagy. In addition, during cGAS-mediated non-canonical autophagy dependent on STING, STING binds WIPI2 through its FRRG motif and competes for binding to PI3P (Wan et al, 2023). The binding of STING to WIPI2, which is essential for LC3 lipidation, is proposed to occur when WIPI2 is bound to membranes. Thus, it would be important to test whether ULK1 phosphorylation at S284 can disrupt STING-dependent autophagy by releasing WIPI2 from the membrane and abolishing the interaction of STING and WIPI2.

Note, we were unable to raise a phospho-specific antibody to S68, and thus it remains a possibility that in cells, S68 is not an ULK1 substrate. However, our western blot data suggest that S68 could be directly phosphorylated by ULK1, or indirectly through an unknown kinase that is itself regulated by ULK1 (Fig. 4B).

Taken together, our data suggest that phosphorylation of WIPI2b at S68 disrupts the WIPI2b-ATG16L1 binding, WIPI2b and ATG16L1 puncta formation, and consequently inhibits starvation-induced autophagy.

## WIPI2b association with membranes is regulated through phosphorylation of S284

Serine 284 is positioned on the unstructured hydrophobic 6CD loop within blade 6, and while S284 is not conserved across all species, there is a serine in the hydrophobic loop in nearly all

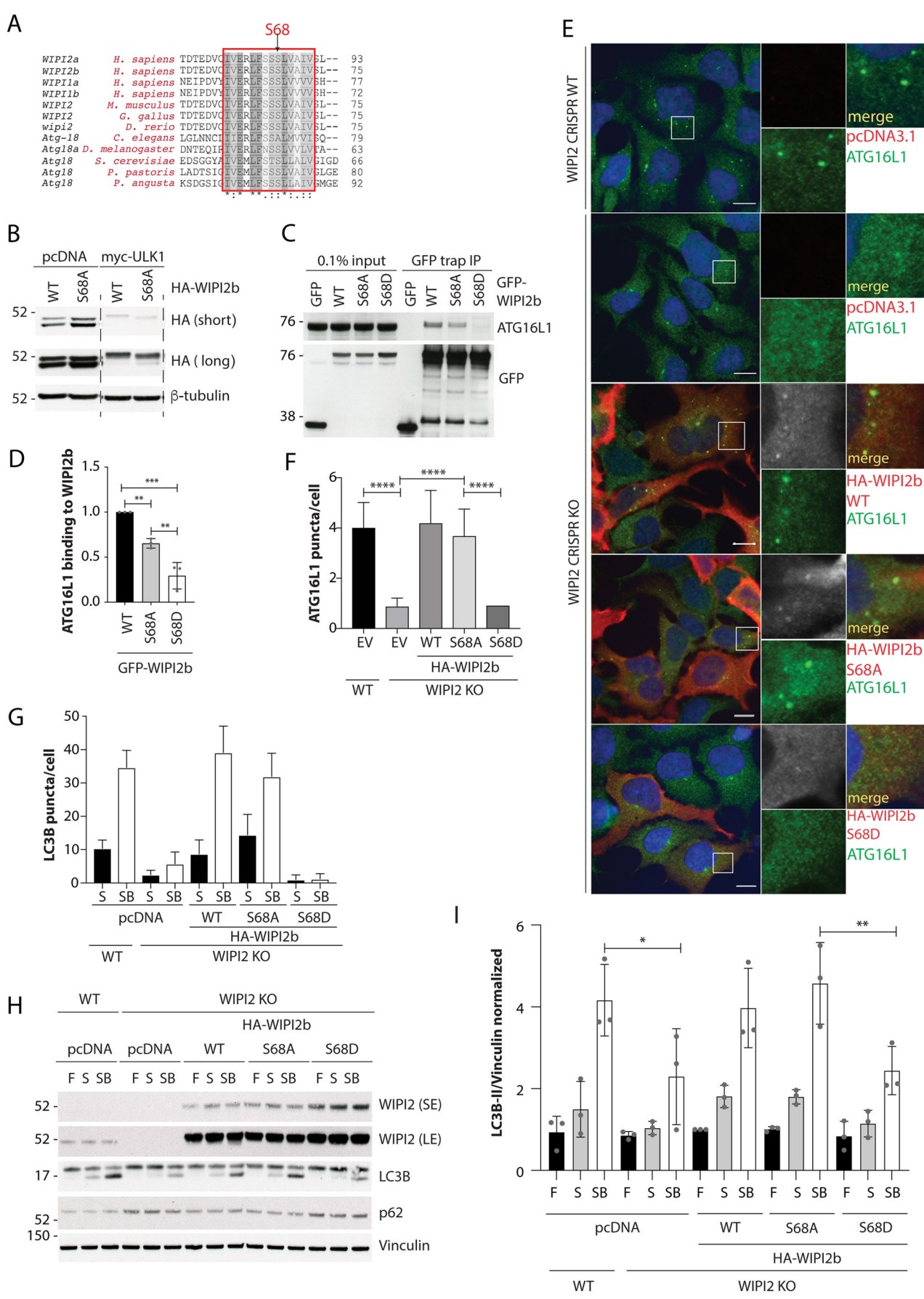

◄ **Figure 4. Phosphorylation at WIPI2b S68 regulates its interaction with ATG16L1.**

(A) Alignment of WIPI2 and homologues reveals high conservation of Serine 68. (B) Co-expression of HA-WIPI2b WT or HA-WIPI2b S68A and either empty vector or myc-ULK1 WT followed by 2 h in EBSS and analysis by western blot. (C) GFP-Trap of lysates from HEK293A cells expressing GFP, GFP-WIPI2b WT, GFP-WIPI2b S68A or GFP-WIPI2b S68D were analysed by western blot for ATG16L1. (D) Quantification of ATG16L1 binding in (C), one-way ANOVA with Tukey's post test. SEM from $n = 3$ biological replicates (2 technical replicates/n). WT vs S68A **$P = 0.0079$, WT vs S68D ***$P = 0.0002$, S68A vs S68D **$P = 0.0066$. (E) WIPI2 WT control cells expressing EV, and WIPI2 CRISPR KO G12 cells expressing EV, HA-WIPI2b WT, HA-WIPI2b S68A and HA-WIPI2b S68D were incubated 2 h in EBSS. ATG16L1 and anti-WIPI2 were visualised by confocal microscopy. Scale bar = 10 μm. (F) Quantification of ATG16L1-positive puncta per cell in (E). SEM from at least 150 cells per condition, $n = 3$ biological replicates. ****$P < 0.0001$ using one-way ANOVA with Tukey's post test. (G) Quantification of LC3-positive puncta in WIPI2 CRISPR KO cells as per Fig. EV4B. SEM from at least 100 cells per condition. $N = 2$ biological replicates. (H) WIPI2 WT cells transiently expressing EV and WIPI2 KO cells expressing EV, HA-WIPI2b WT, HA-WIPI2b S68A or HA-WIPI2b S68D were incubated for 2 h in full medium (F), EBSS (S) or EBSS with Bafilomycin A1 (SB) and analysed by western blot. (I) Quantification of LC3-II levels in (H) was performed by one-way ANOVA with Tukey's post test, *$P = 0.0376$, **$P = 0.01$. SEM from $n = 3$ biological replicates. Source data are available online for this figure.

PROPPINs (Gopaldass et al, 2017). To explore the impact of phosphorylation at S284, we confirmed that WIPI2b was phosphorylated in cells by raising a phospho-specific antibody to S284 (Fig. 5A). Co-expression of WIPI2b with ULK1 led to the appearance of a phosphorylated band recognised by the pS284 antibody. Endogenous IP with the pS284 antibody showed phosphorylation can be detected when cells are treated with okadaic acid (OA) to enrich for phosphorylated proteins (PP1/2A phosphatase inhibitor) (Fig. 5B). Direct phosphorylation of WIPI2b was shown by in vitro kinase assay of IP'ed myc-ULK1 and purified WIPI2b. WIPI2b was phosphorylated when incubated with myc-ULK1 WT but not the kinase-inactive (KI) mutant. Phosphorylation at pS284 was reduced when myc-ULK1 WT and WIPI2b were treated with inhibitors MRT and TORIN1 (Fig. 5D).

Previous work on yeast PROPPINs proposed their association with membranes occurred through sequential activities, in particular, electrostatic interactions followed by insertion of the unstructured, hydrophobic loop into the lipid bilayer, and PI3P binding to stabilise the PROPPIN on the membrane (Busse et al, 2015; Jensen et al, 2022). The structure of HSV2 (Baskaran et al, 2012; Krick et al, 2012; Watanabe et al, 2012) reveals that WIPI2b S284 lies within the unstructured, hydrophobic loop of blade 6, aa 264-296 in WIPI2b, called the 6CD loop. 18 amino acids in the loop are predicted to form an amphipathic α-helix (AH), which has been shown for ATG18 to insert into membranes (Gopaldass et al, 2017). Thus, phosphorylation of S284 could potentially disrupt membrane association. To test this, we fused the AH of WIPI2b (271–288) to a GFP-tagged coiled-coil domain from GMAP210 (39-377) (a bioprobe). GMAP210 coiled-coil (GCC) probe undergoes artificial dimerisation to enhance its membrane affinity (Horchani et al, 2014). We used lipid droplets to monitor membrane association (Horchani et al, 2014; Nishimura and Tooze, 2020). Transiently transfected cells expressing GCC-GFP, WIPI2b WT-GCC-GFP, WIPI2b S284A-GCC-GFP, or WIPI2b S284D-GCC-GFP were treated with oleic acid for 24 h to induce lipid droplets. While the recruitment of WIPI2b WT and S284D bioprobes closely resembled the negative control, WIPI2b S284A showed an increased association with lipid droplets (Fig. EV4A).

As WIPI2b S284D failed to rescue LC3 lipidation in WIPI2 KO cells compared to WIPI2b WT in fed and starved conditions (Fig. 5E), we looked at WIPI2b S284 membrane association by puncta formation (Fig. 5F,G). We observed a reduced number of WIPI2b S284D puncta in starved +/− Baf A1 conditions, compared to WIPI2b WT, while WIPI2b S284A formed more puncta in all conditions. Interestingly, compared to WT and WIPI2 S284D, WIPI2b S284A puncta number was significantly increased

during starvation with the addition of Baf A1. Further investigation into this observation is required to determine where S284A is binding in the presence of Baf A1, but by western blot analysis we do not see a decrease in WT WIPI2 levels over 2 h of starvation (Fig. EV4B,C) so its unlikely to be captured in lysosomes but this remains to be tested.

To understand if these changes were due to effects on association of the mutants to membranes, we performed crude cell membrane fractionation on WIPI2 KO cells overexpressing the WIPI2b WT or S284 phosphorylation mutants (Fig. EV4D,E). There was a significant increase of WIPI2b S284A in the membrane fraction compared to WT and a significant decrease of the S284D. This suggests phosphorylation in the 6CD loop regulates the membrane association of WIPI2b.

To understand the mechanism of membrane association, we next asked if the formation of the AH was affected by phosphorylation at S284. We generated 32 amino acid-long peptides corresponding to the 6CD loop of WIPI2b (aa 264–296), including the WT sequence, and peptides with the S284A and S284D mutations, or phosphorylated at S284 (Fig. EV4F). Gopaldass et al (Gopaldass et al, 2017) used a scrambled AH mutant of Atg18 (yeast homologue of WIPI2) that was unable to form an AH in the presence of liposomes. So, in addition to the phosphorylation mutants, we designed a scrambled mutant (Sloop) for WIPI2b where two pairs of hydrophobic/hydrophilic amino acids within the 6CD loop were swapped or substituted to alter the pattern of hydrophobicity while minimising changes on the overall sequence and hydrophobicity compared to the WT. To detect any changes to the secondary structure between these peptides that may impair membrane binding we collected CD (circular dichroism) spectra for each peptide in the presence or absence of extruded DOPC/DOPE liposomes (100 nm diameter). As expected, the WT peptide was unfolded in aqueous solution and adopted alpha-helical structure in the presence of liposomes. This was also seen for S284A (Figs. 5H,I and EV4G). The Sloop showed no alpha-helical structure with or without liposomes, suggesting the scrambled sequence prevented alpha-helix formation. Interestingly, the S284D and phosphorylated peptide also remained unstructured even when liposomes were added. S284D, the phosphorylated peptide, and Sloop peptide had significantly reduced percentage helicity compared to WT and S284A in the presence of liposomes, with insignificant differences between the WT and S284A. These results suggest that phosphorylation at S284 prevents WIPI2b membrane association by blocking amphipathic α-helix formation. These data further validate our use of WIPI2b S284A and S284D as phospho-null/mimic mutants in cells as the peptides form similar secondary structures to the

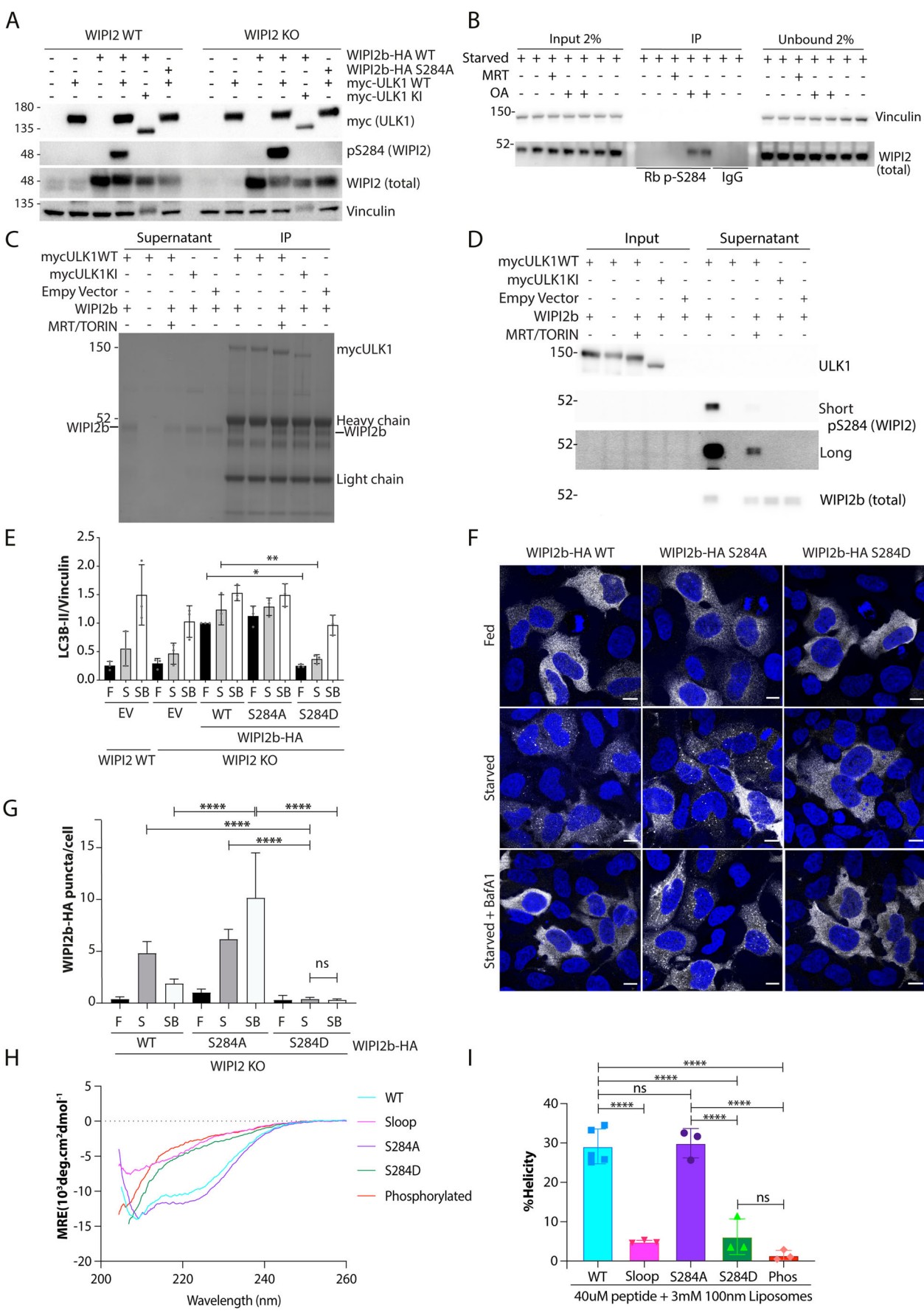

Figure 5.  Phosphorylation at WIPI2b S284 regulates its association with membranes.

(A) WIPI2 CRISPR WT or KO cells were transfected with WIPI2b-HA WT, WIPI2b-HA S284A, myc-ULK1 WT or myc-ULK1 KI, lysed and subjected to western blotting with phospho-specific WIPI2 S284 antibody. Representative experiment from $n = 3$. (B) Endogenous phosphorylated WIPI2 was immunoprecipitated with anti-pS284 antibody crosslinked to protein A (Pt A) sepharose beads with DMP. Rabbit IgG crosslinked to Pt A sepharose beads was the negative control. HEK293A-WT cells were starved for 1 h in EBSS, with 1 μm MRT68921 or 1 μm Okadaic acid (OA). $N = 1$. (C) In vitro Kinase assay with myc-ULK1 WT or KI and purified WIPI2b. The mixed beads were incubated with ATP. Input (cell lysate), supernatant and IP samples were analysed by SDS-PAGE and visualised by Instant blue stain. Representative experiment of $n = 2$. (D) Western blot of (C) with anti-pS284, anti-ULK1 and anti-WIPI2. (E) WIPI2 WT cells expressing empty vector (EV) and WIPI KO cells expressing EV, WIPI2b-HA WT, WIPI2b-HA S284A or WIPI2b-HA S284D were incubated for 2 h in full medium (F), EBSS (S) or EBSS with Bafilomycin A (SB) and analysed for LC3 lipidation by western blot. Mean ± SEM from $n = 3$ biological replicates, one-way ANOVA with Tukey's post test, $*P = 0.014$, $**P = 00026$. (F) WIPI2 CRISPR KO cells transiently expressing WIPI2b-HA WT, WIPI2b-HA S284A or WIPI2b-HA S284D were incubated 2 h in EBSS (Starved), EBSS with Bafilomycin A1 or left untreated (Fed). The cells were analysed by confocal microscopy after immunostaining with anti-HA antibody. Scale bar = 10 μm. (G) Statistical analysis of HA-positive puncta in (F) SEM from at least 100 cells per condition. $N = 3$ biological replicates. $****P < 0.0001$ using one-way ANOVA with Tukey's post test. (H) Far UV CD spectra of 40 μM WT and mutant WIPI2b amphipathic helix peptides in the presence of 3 mM 100 nm liposomes (DOPC/DOPE 70/30). Raw CD data were converted to MRE (mean residue ellipticity). (I) Percentage helicity calculated from MRE at 222 nm shown in (F). Means with SD, $n = 3$ biological replicates, one-way ANOVA and Tukey's multiple comparison test. $****P < 0.0001$. Source data are available online for this figure.

WT/phosphorylated forms. Our CD data suggests phosphorylation causes secondary structure changes within the 6CD loop, likely destabilising WIPI2b at the phagophore membrane and regulating its function.

Finally, we wanted to confirm that the phosphorylation of WIPI2b at S284 does not affect its binding to ATG16L1 or ULK1. To test if there was a disruption of binding between WIPI2b S284D and ATG16L1, we expressed and immunoprecipitated GFP-WIPI2b WT, S284A and S284D, alongside to S68A and S68D, and tested the interaction with ATG16L1. We observed no significant difference in ATG16L1 binding between WIPI2b WT, S68A, S284A and S284D, while S68D binding to ATG16L1 was, as expected, reduced. (Fig. EV4H,I). WIPI2b S284A and S284D binding to ULK1 and ATG13 also remained unchanged (Fig. EV4H,J). Taken together, these data show that WIPI2b is phosphorylated by ULK1 at S284, and that phosphorylation regulates the formation of the AH in the 6CD loop, and subsequently the association of WIPI2b with membranes, and mutation of S284 to alanine appears to enhance membrane binding.

A possible explanation for the impact of phosphorylation of S284 on localisation and autophagy could be that WIPI2b's lipid specificity is altered by the S284A mutation, meaning S284A interacts more strongly with another lipid species compared to the WT. *S. cerevisiae* Atg18 contains a stretch of 18 amino acids within the 6CD loop, which upon contact with membranes, folds into an AH (Gopaldass et al, 2017), and it is predicted that the properties of this stretch of amino acids is conserved in other PROPPINs, including WIPI2. The predicted WIPI2d AH formed stable interactions with membranes, promoting membrane recruitment of ATG16L1 (Jensen et al, 2022). Of note, while the hydrophobic loop and the unstructured C-terminus of WIPI2b/d were excluded from the protein used in the crystal structure by Strong et al (Strong et al, 2021), a predicted model of WIPI2 using Alphafold contains the hydrophobic loop, and, additionally, it predicts that the stretch of 18 amino acids indeed folds into a helix, albeit with low confidence. Thus, one mechanism for regulating WIPI2b membrane association would be phosphorylation at S284, which alters the hydrophobicity of the loop, causes WIPI2b to dissociate from membranes or prevents WIPI2b to re-associate with membranes.

Together, our work provides further mechanistic understanding of how the function of WIPI2b is regulated at the phagophore. The working model is as follows (Fig. EV5); WIPI2b is recruited to the phagophore. WIPI2b binds ATG16L1, directing LC3 and GABARAP lipidation. ULK1 binds WIPI2b and phosphorylates it at S68 and S284. The sequence of these phosphorylation events remain unknown. We hypothesise S68 is phosphorylated first, disrupting ATG16L1 binding, and potentially destabilising WIPI2b at the membrane. This may allow ULK1 to access the 6CD loop and phosphorylate WIPI2b at S284, preventing amphipathic helix formation and allowing WIPI2b to dissociate from the phagophore so later stages of autophagosome biogenesis can take place.

One of the outstanding questions is the role of phosphatases in the regulation of WIPI2 activity, especially given that WIPI2b binds ATG16L1 and PI3P-positive membranes only when dephosphorylated. Protein Phosphatase 2A (PP2A) has already been implicated in the dephosphorylation of major autophagy-related proteins (Pengo et al, 2017; Wong et al, 2015). PP2A dephosphorylates ULK1 at S637 (Wong et al, 2015) and ATG4B at Serine 316 (Pengo et al, 2017) upon amino acid starvation-inducing autophagy, and controlling the processing of LC3. Further work is needed to identify the specific phosphatase involved in the kinetics of WIPI2 phosphorylation and dephosphorylation but our work showing the effect of okadaic acid supports this possibility, at least for S284. Understanding this intricate process at the phagophore is crucial, as it has previously been highlighted as a potential therapeutic target for addressing age-related neurological conditions (Stavoe et al, 2019).

## Methods

### Cell culture, DNA plasmids and antibodies

HEK293A and its derivatives were cultured in DMEM (Dulbecco's modified Eagle's medium) supplemented with 10% foetal bovine serum (FBS) and 4.8 mM L-glutamine. Cells were washed and incubated in Earle's balanced salt medium (EBSS) to induce autophagy by amino acid starvation. Where indicated, cells were treated with 100 nM Bafilomycin or 100 nM Wortmannin for two h, with 1 μM MRT68921,100 nM SBI-0206965, 1 μM okadaic acid for 1 h, or with 1 mM oleic acid for 24 h.

WIPI2 KO cell lines were generated by CRISPR/Cas9. Guide RNAs were designed using the Broad Institute CRISPR tool provided by the F. Zhang laboratory (Broad Institute, MIT, Boston, MA). A selected target sequence was in the third exon, which is the first exon conserved across other WIPI2 isoforms (clone G12C11 sgRNA WIPI2, forward 5'-CACCGCAGCTACTCCAACACGATTC-3'; reverse 5'- AAAC GAATCGTGTTGGAGTAGCTGC-3'; clone G41 sgRNA WIPI2,

**Table 1. DNA constructs used in this study.**

| Name | Vector | Source |
|------|--------|--------|
| pcDNA3.1+ | pcDNA3.1+ | ThermoFisher |
| pEGFPC1 | pEGFPC1 | Clontech |
| GFP-WIPI2b WT | pEGFPC1 | Polson et al, 2010 |
| GFP-WIPI2b FTTG | pEGFPC1 | Polson et al, 2010 |
| GFP-WIPI2b RERE | pEGFPC1 | Dooley et al, 2014 |
| GFP-WIPI2b S68A | pEGFPC1 | This study |
| GFP-WIPI2b S68D | pEGFPC1 | This study |
| GFP-WIPI2b S284A | pEGFPC1 | This study |
| GFP-WIPI2b S284D | pEGFPC1 | This study |
| HA-WIPI2b WT | pDest HA | This study |
| HA-WIPI2b S68A | pDest HA | This study |
| HA-WIPI2b S68D | pDest HA | This study |
| WIPI2b-HA WT | pcDNA3.1 | This study |
| WIPI2b-HA S68A | pcDNA3.1 | This study |
| WIPI2b-HA S68D | pcDNA3.1 | This study |
| WIPI2b-HA S284A | pcDNA3.1 | This study |
| WIPI2b-HA S284D | pcDNA3.1 | This study |
| WIPI2b-HA S395A | pcDNA3.2 | This study |
| WIPI2b-HA S395D | pcDNA3.2 | This study |
| WIPI2b-HA 6A | pcDNA3.1 | This study |
| WIPI2b-HA 6D | pcDNA3.1 | This study |
| WIPI2b-HA 7A | pcDNA3.1 | This study |
| myc-WIPI2b WT | pCMVTag3 | Polson et al, 2010 |
| myc-mULK1 WT | pRK5 Myc | Tomoda et al, 1999 |
| myc-mULK1 KI | pRK5 Myc | Chan et al, 2009 |
| HA-hULK1 WT | pcDNA3.1+ | Longatti et al, 2012 |
| HA-hULK1 KI | pcDNA3.1+ | This study |
| pEGFP N1: GCC (GMAP210 (39-377)) | pEGFP N1 | Habib Horchani et al, 2014 |
| pEGFP N1: hWIPI2b (271–288 aa) WT-GCC (GMAP210 (39-377)) | pEGFPN2 | This study |
| pEGFP N1: hWIPI2b (271–288 aa) S284A-GCC (GMAP210 (39-377)) | pEGFPN3 | This study |
| pEGFP N1: hWIPI2b (271–288 aa) S284D-GCC (GMAP210 (39-377)) | pEGFPN4 | This study |
| 6XHis-MBP-GST-3C-WIPI2b | pBacPak | This study |

forward 5'- CACCGAATGCACACATCTTCCGTAT-3'; reverse 5'-AAACATACGGAAGATGTGTGCATTC-3'). The appropriate oligonucleotides were aligned and cloned into BbsI site of pSpCas9(BB)-2A-GFP (F. Zhang, MIT, Boston, MA; Addgene 48138), according to the protocol provided by the F. Zhang laboratory. The successfully cloned plasmid was transfected into HEK293A cells, and GFP-positive cells were selected by flow cytometry. Clones were obtained by serial dilution and screened for WIPI2 levels by western blot and immunofluorescence.

For transient transfection of cells, Lipofectamine 2000 (Invitrogen) was used according to the manufacturer's instructions. DNA plasmids were used at a concentration of 1 µg/ml of transfection mix. Where indicated, pcDNA3.1+ and pEGFPC1 were used as a vector control. GFP-WIPI2b WT, FTTG and RERE, myc-WIPI2b WT (Dooley et al, 2014), WIPI2b-HA WT (Gilad and Kimchi, 2014), myc-mULK1 WT (Tomoda et al, 1999), myc-mULK1 K46I (Chan et al, 2009), HA-hULK1 WT (Kraft et al, 2012) were previously described. HA-WIPI2b WT was cloned using Gateway cloning technology according to the manufacturer's instructions, into pDest HA, previously described (Lamark et al, 2003). GFP-WIPI2b, HA-WIPI2b and WIPI2b-HA phospho- and non-phosphomutants were generated using QuikChange Multi Site-Directed Mutagenesis kit (Agilent Technologies). mCherry-ATG13 was a kind gift from N. Ktistakis (Babraham Institute, Cambridge, UK) (Karanasios et al, 2013). 3xflag-FIP200 was a kind gift from N. Mizushima (University of Tokyo, Tokyo, Japan) (Hara et al, 2008). pEGFP N1:GCC$_{GMAP210 (39-377)}$ was a kind gift from B. Anthony (Université Côte d'Azur, France) (Magdeleine et al, 2016). pEGFP N1: WIPI2b $_{GMAP210 (39-377)}$ constructs (WT, S284A and S284D) were generated using conventional cloning. The list of plasmids used in this study can be found in Table 1.

Mouse antibodies: anti-WIPI2 (Polson et al), anti-ATG16L1 (for WB) (MBL, M150-3), anti-Vinculin (Sigma, V9264), anti-p62 (BD Biosciences, 610832), anti-FLAG M2 (Sigma), anti-GFP (CRUK, 3E1), anti-HA.11 (Biolegend, 901501), anti-myc (CRUK, 9E10). Rabbit antibodies: anti-ATG16L1 (for WB and IF) (MBL, PM040), anti-ULK1 (for WB, Santa Cruz, sc-33182; for WB and IF, Cell Signaling, 8054 D8H5), anti-ATG13 (Sigma, SAB4200100), anti-LC3B for WB (Abcam, ab48394), anti-Actin (Abcam, ab8227), anti-β-Tubulin (Abcam, ab6046), anti-GFP (Santa Cruz, sc-8334), anti-RFP (MBL, PM005), anti-SOD1 (Abcam, ab16831), anti-Vamp3 (Thermo Fischer, PA1-767A) were used at the manufacturer's suggested concentrations. Rabbit polyclonal antibodies (WIPI2 pS284) were raised in-house using phosphorylated peptides. 15mer peptides centred on WIPI2 serine 284 encoding a phosphoserine in the central position ((C)FGKVLMA(Sp)TSYLPS) were generated in-house and used to immunise rabbits at Covalab (FR). The antibodies were affinity-purified with the phospho-S284 peptide. Secondary antibodies for IF, from Life Technologies unless otherwise specified, were anti-rabbit IgG Alexa Fluor 488, 555 and 647, anti-mouse IgG Alexa Fluor 488 and 555. HRP-conjugated secondary antibodies used for WB were from GE Healthcare.

## Western blotting

Cells were lysed using ice-cold TNTE lysis buffer (20 mM Tris-HCl pH 7.5, 150 mM NaCl, 5 mM EDTA, 0.3% Triton-X), supplemented with 1x Complete protease inhibitor cocktail (Roche) and 1 × PhosSTOP (Roche). EDTA and PhosSTOP were not included in samples treated with Lambda phosphatase. Lysates were cleared by centrifugation. For dephosphorylation of WIPI2, the lysate was incubated with 1 µl of lambda phosphatase (NEB, P0753) with the accompanying manufacturer's buffer and 1 mM MnCl$_2$. Both buffer and MnCl$_2$ were included in control samples, but instead of lambda phosphatase, Sodium Orthovandate was added at the final concentration of 100 µM. The reaction was carried out for 30 min at 30 °C. 5 × Laemmli sample buffer was added to samples to stop the reaction. Proteins were resolved on Nu-Page Bis-Tris 4–12%

gels (Life Technologies) or self-made 8% Tris-Glycine gels for protein mobility shifts. The proteins were then transferred onto a PVDF membrane (Merck Millipore). Following incubation with primary and secondary antibodies, the blots were developed by enhanced chemiluminescence (GE Healthcare). pS284 antibody was diluted in Solution 1 (Millipore signal boost immunoreaction enhancer for primary antibodies) for optimum signal. Densitometry was performed with ImageJ software. For western blotting of antibodies exhibiting weak signal, blots were developed with Luminata Crescendo Western HRP substrate (Merck Millipore).

## Crosslinking antibody to sepharose beads

In total, 20 µg of WIPI2 antibody was incubated with 30 µl of Protein (Pt) G or A beads in PBS, rotating overnight at 4 °C. The beads were washed three times in 0.2 M sodium borate, pH 9, before their incubation in 20 mM DMP in 0.2 M sodium borate, pH 9 suspension, rotating at room temperature for 40 min. The beads were washed in 0.2 M ethanolamine, pH 8, and then incubated in the same solution for 1–2 h at room temperature. Beads were then washed three times in 0.58% v/v acetic acid with 150 mM NaCl, followed by three washed in 1 ml of cold PBS. The beads were resuspended in 50 µl PBS and ready to be used in immunoprecipitation experiments.

## Immunoprecipitation

Prior to immunoprecipitation (IP) of endogenous protein, anti-WIPI2 or anti-pS284 antibodies were crosslinked to Protein G (Pt G) or Protein A (Pt A) beads using DMP (dimethylpimelimidate) crosslinker. Cells were lysed in TNTE lysis buffer, and the lysate was subjected to incubation with either WIPI2-Pt G overnight, or GFP-trap beads for two h, rotating at 4 °C. All GFP Trap IP samples were washed in TNTE wash buffer (50 mM Tris pH 7.5, 300 mM NaCl, 5 mM EDTA, 0,1% Triton-X100, 0.02% NaN$_3$), and endogenous IP beads were washed in low-stringency lysis buffer (150 mM NaCl, 50 mM Tris-HCl pH 7.4, 10 mM EGTA, 0.2% NP-40). Proteins were eluted with 2x Laemmli sample buffer at 100 °C for 5 min before resolving by SDS-PAGE and western blotting.

## Immunofluorescence

Cells were grown on coverslips, fixed with 3% paraformaldehyde for at least 20 min and permeabilised with either methanol (room temperature; for LC3 and HA staining) or 0.1% Triton-X100 (for ATG16L1 staining) for 5 min. Coverslips were blocked in 5% BSA (Roche) for one h, incubated in primary antibody in 1% BSA for 1 h, washed three times in PBS, incubated in secondary antibody and Hoechst (final concentration 5 µg/ml) in 1% BSA for one h, before final washes with PBS and water. LC3, ATG16L1 and WIPI2b puncta formation was assessed by Imaris image analysis software.

## In vitro kinase assay

HEK293A cells transiently expressing the kinase (myc-ULK1) were starved in EBSS for 1 h to induce autophagy and activate the kinase or treated with EBSS containing 1 µM MRT68921 and 100 nm TORIN1. The cells expressing WIPI2b-HA were left untreated. The cells were lysed and IP of myc-ULK1 or WIPI2b-HA was performed using myc-Trap beads (Chromotek) or HA-affinity matrix (Roche), respectively.

The beads were then washed once and then resuspended in kinase reaction buffer (20 mM HEPES pH 7.5, 20 mM MgCl$_2$, 25 mM β-glycerophosphate, 2 mM DTT, 100 µM Sodium Orthovanadate). myc-ULK1 and WIPI2b-HA immunoprecipitates were mixed for the assay or left by its own as control. When using purified WIPI2b, 5 µg was used per reaction. The beads were then incubated in 32 µl of kinase buffer containing 20 µM ATP and 2 µC [γ-$^{32}$P] ATP for 30 min at 30 °C. In total, 5 × Laemmli sample buffer was then added to each sample to stop the kinase reaction. The samples were boiled for 5 min before resolving on 8% hand-poured SDS-PAGE gel. The gel was then incubated in Instant Blue dye (Sigma) for one h and destained overnight in water. The gel was dried in Laboratory Gel Dryer (Bio-Rad) and visualised by exposing to film (Amersham Hyperfilm ECL, GE Healthcare). For mass spectrometry (myc-ULK1 + purified WIPI2b) analysis, [γ-$^{32}$P] ATP was omitted. After the incubation the beads were spun down and the supernatant collected. For validation of the IP, and to detect the pS284, 5 × Laemmli sample buffer was added to a small aliquot to run on a Nu-Page Bis-Tris 4–12% gels (Life Technologies) gel for Instant Blue staining and western blot analysis, and the remainder was processed for mass spectroscopy.

## Crude cell fractionation

HEK293A cells and its derivatives were washed once in buffer containing 20 mM HEPES-KOH, pH 7.5, 10 mM KCl, 2.5 mM MgOAc and 1 mM EDTA, and incubated on ice for 10 min. Upon removal of wash buffer, the cells were collected in buffer containing 20 mM HEPES-KOH, pH 7.5, 2.5 mM MgOAc, 1 mM EDTA, 250 mM sucrose, 1 mM dithiothreitol (DTT), 1 × PhosSTOP (Roche) and 1 × Complete protease inhibitors cocktail (Roche). Cells were then homogenised by passing through a 27 G needle, and homogenisation was monitored by Trypan Blue staining. Nuclei were removed by centrifugation at 1000 × g for 5 min at 4 °C. This was repeated, and the cleared supernatant was then subjected to centrifugation at 100,000 × g for 1 h at 4 °C. The obtained pellet contained membranes, while the supernatant produced the true cytosol. The supernatant was transferred to a clean Eppendorf tube and 5 × Laemmli sample buffer was added. The membrane pellet was resuspended in 2 × Laemmli sample buffer to either the same volume as the cytosol (Fig. EV4D) or a 1/5th of the volume (Fig. EV2F). All samples were heated for 10 min at 65 °C and processed by western blotting.

## Mass spectrometry

HEK293A cells transiently expressing pcDNA3.1+ or GFP-WIPI2b WT with either HA-ULK1 WT or HA-ULK1 K46I, were lysed in TNTE lysis buffer as described above. The lysates were clarified by centrifugation. The lysates expressing GFP-WIPI2b and HA-ULK1 WT/K46I were incubated with GFP-Trap beads overnight for maximum binding efficiency, while the lysates from cells expressing empty vector and HA-ULK1 WT/K46I were incubated overnight with WIPI2 antibody crosslinked to Pt G beads, as described above. Proteins were eluted from the beads in 2 × Laemmli sample buffer at 100 °C for 5 min.

Eluted proteins were separated by SDS-PAGE, and the gel was stained in GelCode Blue Stain Reagent (Thermo Scientific). The bands corresponding GFP-WIPI2b or WIPI2 were cut out of the gel. Gel pieces were processed using in-gel trypsin digestion procedure adapted for a Janus liquid handling system (Perkin Elmer). Alternatively, the

supernatant fraction from in vitro kinase IP with purified WIPI2b and myc-ULK1 was processed directly using trypsin digestion. Peptide extracts were dried to completion using vacuum centrifugation. Peptides were reconstituted in 0.1% Trifluoroacetic acid and resolved by liquid chromatography system. Data was acquired using Orbitrap Fusion Lumos Tribrid mass spectrometer (Thermo Scientific). All raw files were first converted to MGF files and analysed by MASCOT software (Matrix Science), queried against the Uniprot human database to identify peptide matches. Raw files and MASCOT search were then imported into Skyline/Scaffold software and used to visualise the peak areas for the peptides of interest.

## Circular dichroism

Peptides of the WIPI2b amphipathic helix WT and mutant sequence:

WIPI2b WT KPPEEPTTWTGYFGKVLMASTSYLPSQVTEMF;
WIPI2b Sloop KPPEEPTTKTGYQGWVLFASTSYLPSQVTEMF;
S284A KPPEEPTTWTGYFGKVLMAATSYLPSQVTEMF;
S284D KPPEEPTTWTGYFGKVLMADTSYLPSQVTEMF;
phosphorylated KPPEEPTTWTGYFGKVLMAS(P)TSYLPSQV TEMF,

were synthesised by the Francis Crick Peptide Chemistry Technology Platform. Peptide concentrations were determined by measuring the absorbance at 280 nm. Lipids DOPC and DOPE were mixed in a 70/30 ratio and dried by nitrogen gas. They were placed in a vacuum to remove the organic solvent. Buffer A (50 mM NaCl, 20 mM $NaHPO_4$ pH 6.8, 1 mM $MgCl_2$) was used to rehydrate and resuspend the lipid films. Five freeze-thaw cycles were performed using liquid nitrogen and a heat block at 65 °C. Using a mini-Extruder (Avanti Polar Lipid) the suspension was extruded 10.5 times through a 200 nm and 20.5 times through a 100 nm Whatman membrane. Liposome size was confirmed by Zetasizer nano ZS (Malvern Instruments). For circular dichroism (CD) measurements peptides were diluted to a final concentration of 40 μM in buffer A alone or mixed with 3 mM 100 nm liposomes. CD spectra were recorded at 20 °C in a 1 mm path length cuvette using a Jasco J-815 spectropolarimeter, fitted with a cell holder thermostatted by a CDF-426S Peltier unit. A baseline for the buffer was acquired and subtracted from measurements. Each spectrum is the average of 25 scans from 190 to 260 nm, with a bandwidth of 2 nm and a scan speed of 200 nm $min^{-1}$. Raw CD data were converted to molar ellipticity, (Martin and Schilstra, 2008) and percentage helicity was calculated from the molar ellipticity at 222 nm $[(\partial)_{222nm}]$ according to % α-helix = $[(\partial)_{222nm} + 2340]/303$ (Horchani et al, 2014).

## Expression of WIPI2b

The 6XHis-MBP-GST-3C-WIPI2b coding sequence was assembled by restriction enzyme cloning. 0.5μg of DNA was used to transfect $0.5 \times 10^6$ SF9 cells using the FuGene transfection reagent (FuGene HD; Promega). SF900-II medium containing Gentamycin and Amphotericin B was used to select for transfected cells and virus at P0 was harvested to produce a P1 stock. P1 was used to infect 50 mL of culture at a density of $1 \times 10^6$ cells/mL. Cells were pelleted at $1000 \times g$ 72 h after infection and the supernatant taken for P2 virus stock. In total, 50 μl P2 was used to infect 50 mL of culture for ~48 h when cells were pelleted at $1000 \times g$ to collect P3. 1 mL of P3 virus was sued to infect 400 mL culture at a density of $1.8 \times 10^6$ cells/mL for ~48 h before pelleting at $1000 \times g$ and flash frozen in liquid nitrogen and stored at −80 °C until purification.

## Purification of WIPI2b

Pellets were resuspended in 50 mL lysis buffer (50 mM Tris 7.5, 500 mM NaCl, 0.5 mM TCEP, 10% glycerol, 2 mM $MgCl_2$, 0.5% Triton-X, 1.5 × Protease inhibitor cocktail). Lysate was sonicated 3–4 × 10s at 4 °C. In total, 5 μl Benzonase nuclease was added and incubated on ice for 10 min. Lysates were cleared at $30,000 \times g$ for 30 min, 4 °C. 2 mL slurry of Glutathione Sepharose 4B 10324511 (Cytiva) beads were taken and washed 3 × in lysis buffer. Beads were added to cleared lysates rotated for 2 h, 4 °C. Beads were separated at $200 \times g$, and supernatant taken as unbound. Beads were washed 5 × in wash buffer (50 mM Tris 7.5, 500 mM NaCl, 0.5 mM TCEP, 10% glycerol, 2 mM $MgCl_2$,). Beads were resuspended in 15 mL wash buffer and GST-3C protease added and rotated overnight at 4 °C in a 15 mL Eppendorf. The elution was collected and concentrated in a 20 mL 10 K Vivaspin V2002 concentrator to 0.5 mL. The sample was run on a superdex200 column for size exclusion chromatography in equilibration buffer (25 mM Tris-HCl pH 7.5, 150 mM NaCl, 0.5 mM TCEP, 10% Glycerol, 1 mM $MgCl_2$). Fractions corresponding to the monomeric peak were pooled together and concentrated to 1–2 mg/ml.

## Statistical analysis

Statistics were performed using GraphPad Prism 7 software, as detailed in the figure legends.

# Data availability

This study includes no data deposited in external repositories.

The source data of this paper are collected in the following database record: biostudies:S-SCDT-10_1038-S44319-024-00215-5.

# Peer review information

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

## Acknowledgements

The authors thank past and present members of the MCBA lab for helpful comments and support. We acknowledge the excellent technical assistance from the Francis Crick Institute Science Technology Platforms, in particular Structural Biology, Proteomics, and Peptide Synthesis. The authors thank Bruno Antony for a cDNA coding a coiled-coil domain from GMAP210 (39-377). The authors thank Estelle Descamps for preliminary experimental results. This work was supported by the Francis Crick Institute, which receives its core funding from Cancer Research UK (CC2134 and CC1068); the UK Medical Research Council (CC2134 and CC1068); and the Wellcome Trust (CC2134 and CC1068).

## Author contributions

**Andrea Gubas**: Conceptualisation; Data curation; Formal analysis; Validation; Investigation; Methodology; Writing—original draft; Writing—review and editing. **Eleanor Attridge**: Data curation; Formal analysis; Validation; Investigation; Methodology; Writing—review and editing. **Harold BJ Jefferies**: Data curation; Validation; Investigation; Methodology. **Taki Nishimura**: Data curation; Investigation; Methodology; Writing—review and editing. **Minoo Razi**: Data curation; Investigation. **Simone Kunzelmann**: Data curation; Writing—review and editing. **Yuval Gilad**: Data curation. **Thomas J Mercer**: Validation; Investigation; Methodology; Writing—review and editing. **Michael M Wilson**: Data curation; Software; Investigation; Visualisation. **Adi Kimchi**: Data curation; Supervision; Funding acquisition. **Sharon A Tooze**: Conceptualisation; Supervision; Writing—review and editing.

Source data underlying figure panels in this paper may have individual authorship assigned. Where available, figure panel/source data authorship is listed in the following database record: biostudies:S-SCDT-10_1038-S44319-024-00215-5.

## Funding

## Disclosure and competing interests statement

The authors declare no competing interests.

# Expanded View Figures

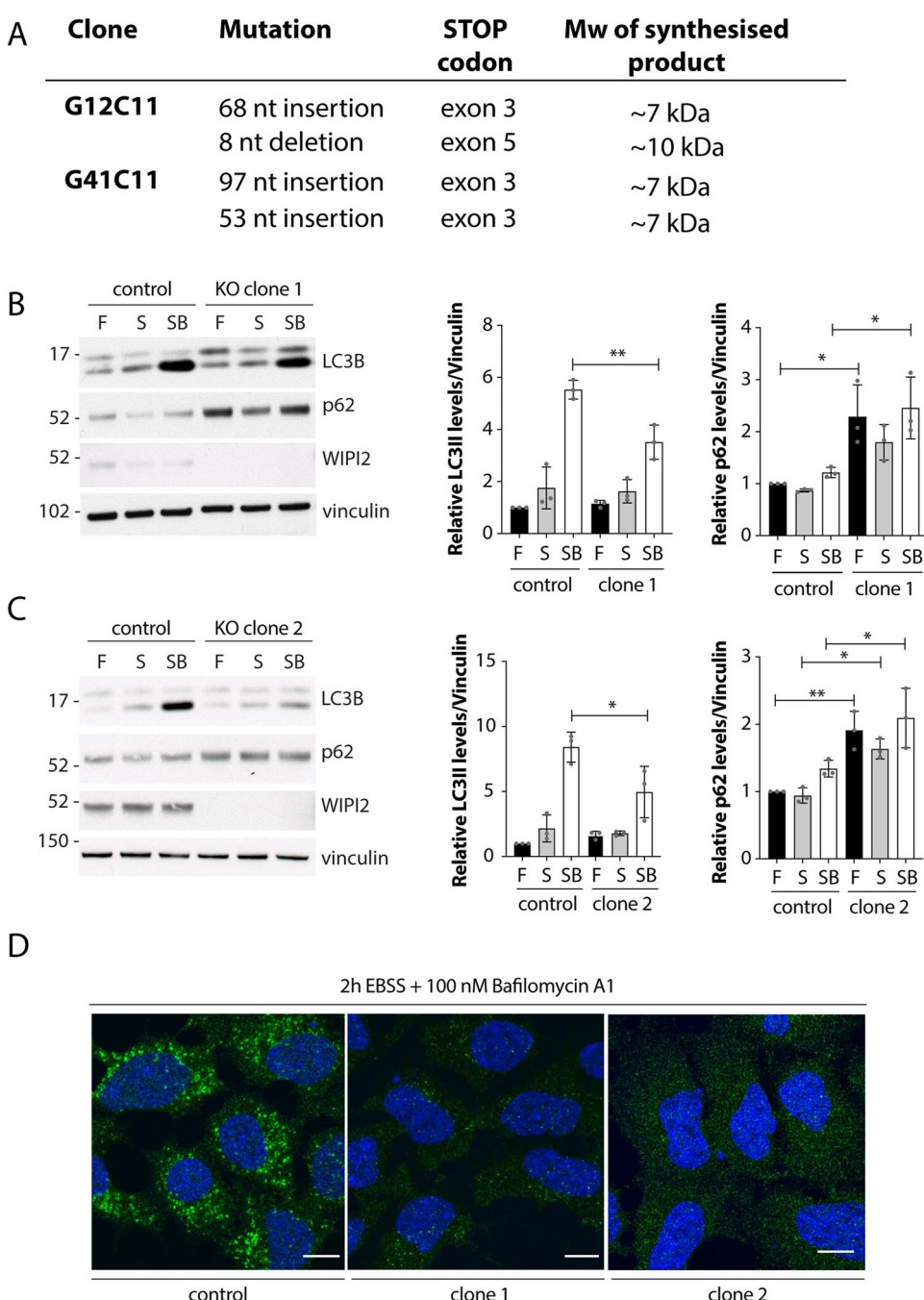

| Clone | Mutation | STOP codon | Mw of synthesised product |
|---|---|---|---|
| **G12C11** | 68 nt insertion | exon 3 | ~7 kDa |
| | 8 nt deletion | exon 5 | ~10 kDa |
| **G41C11** | 97 nt insertion | exon 3 | ~7 kDa |
| | 53 nt insertion | exon 3 | ~7 kDa |

**Figure EV1. Validation of WIPI2KO cells.**

(**A**) WIPI2 CRISPR KO clones and gene mutations. (**B**) WIPI2 control and KO clone G12C11 (clone 1) were incubated 2 h in full medium (F), EBSS (S) or EBSS supplemented with Bafilomycin A1 (SB) and analysed by western blot. LC3-II levels from 3 independent experiments were quantified using one-way ANOVA with Tukey's post test. **P = 0.0029. p62 levels from 3 independent experiments were quantified using one-way ANOVA with Tukey's post test. F vs F *P = 0.0129. (**C**) WIPI2 control and KO clone G41C11 (clone 1) were treated as in (**B**). LC3-II levels from 3 independent experiments were quantified using one-way ANOVA with Tukey's post test. *P = 0.015. p62 levels from 3 independent experiments were quantified using one-way ANOVA with Tukey's post test. F vs F **P = 0.0045, S vs S *P = 0.0312, SB vs SB *P = 0.0183. (**D**) WIPI2 control, G12C11 (clone 1) and G41C11 (clone 2) cell lines were incubated in EBSS with Bafilomycin A1 for 2 h and analysed by confocal microscopy after staining with LC3 antibody. Scale bar = 10 µm.

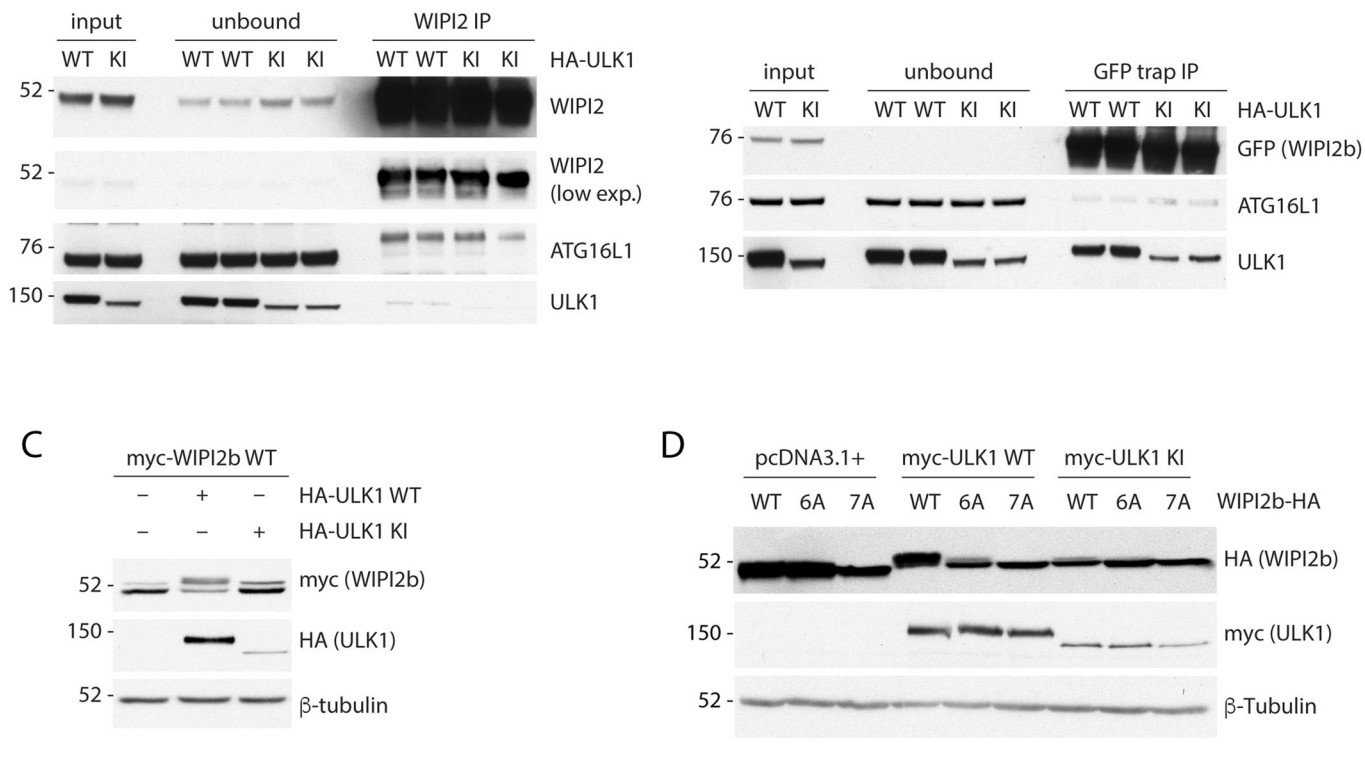

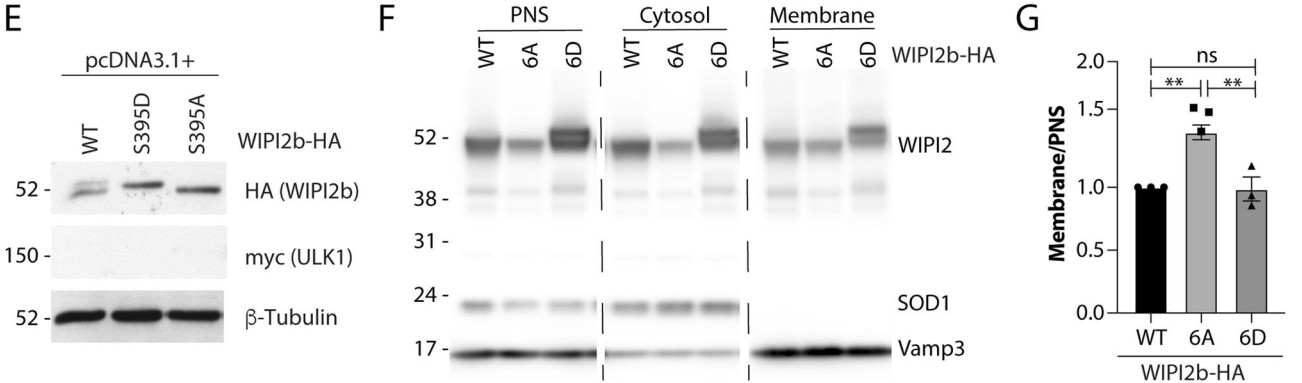

**Figure EV2.  WIPI2b is phosphorylated by ULK1 at 6 sites.**

(A, B) Samples used for mass spectrometry for detection of phosphorylation sites on endogenous WIPI2 (A) and transiently expressed GFP-WIPI2b WT (B) with transient expression of HA-ULK1 WT or KI before and after immunoprecipitation were analysed by SDS-PAGE and western blot. (C) HEK293A cells were transiently transfected with empty vector, HA-ULK1 WT or HA-ULK1 KI and myc-WIPI2b WT and starved for 2 h in EBSS before being analysed by western blot. (D) HEK293A cells transiently co-expressing WIPI2b-HA WT, WIPI2b-HA 6A or WIPI2b-HA 7A and empty vector, myc-ULK1 WT or myc-ULK1 KI were treated for 2 h in EBSS and analysed by western blot with indicated antibodies. Representative experiment from $n = 3$. (E) HEK293A cells co-expressing WIPI2b-HA WT, WIPI2b-HA S395D or WIPI2b-HA S395A and incubated for 2 h in EBSS, lysed and analysed by western blot. Representative experiment from $n = 3$. (F) Crude cellular fractionation was performed on WIPI2 KO cells transiently expressing WIPI2b-HA WT, 6A or 6D. Membrane fraction was solubilised to a 1/5th of the volume of the cytosol. Equal volumes of each fraction were analysed by western blot. SOD1 was used as a marker for the cytosolic fraction, and Vamp3 as a marker for the membrane fraction. (G) Quantification of WIPI2b-HA membrane levels in (F). Membrane fraction over post-nuclear supernatant input (PNS). Mean with SEM using one-way ANOVA with Tukey's post test from 3 independent experiments. WT vs 6A **$P = 0.0069$ 6 A vs 6D **$P = 0.0060$.

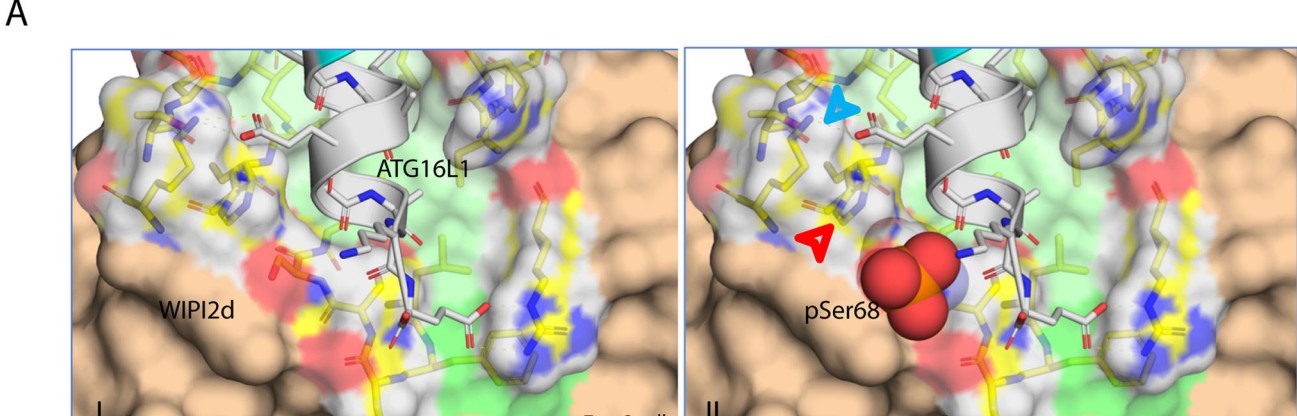

◀ **Figure EV3. Phosphorylation at WIPI2b S68 reduces WIPI2 and LC3 puncta formation.**

(A) (I) Model of human WIPI2d bound to ATG16L1 α-helix (Strong et al, 2021). PDB 7mu2.pdb. Atoms of contact sites coloured as follows – oxygen, red; nitrogen, blue; carbon WIPI2d yellow, ATG16L1 grey; hydrophobic residues shaded green according to hydrophobicity scale (White and Wimley, 1998). (II) Phosphoserine 68 is represented as a red molecule in the bottom image. Residues H85 and K88 are marked by a red and blue arrowhead respectively. (B) WIPI2 control cells expressing empty vector and WIPI2 KO cells expressing empty vector, HA-WIPI2b WT, HA-WIPI2b S68A and HA-WIPI2b S68D were incubated 2 h in EBSS with or without Bafilomycin A1. Cells were analysed by confocal microscopy after immunostaining with LC3B and HA antibodies. Scale bar = 10 μm. (C) WIPI2 KO cells were transiently transfected with HA-WIPI2b WT, HA-WIPI2b S68A and HA-WIPI2b S68D, and starved for two h or left untreated. Cells were analysed by immunostaining with anti-HA antibody. Scale bar = 10 μm. (D) Quantification of (C) HA-positive puncta. SEM from at least 120 cells per condition from two experiments. One-way ANOVA with Tukey's post test ****$P < 0.0001$.

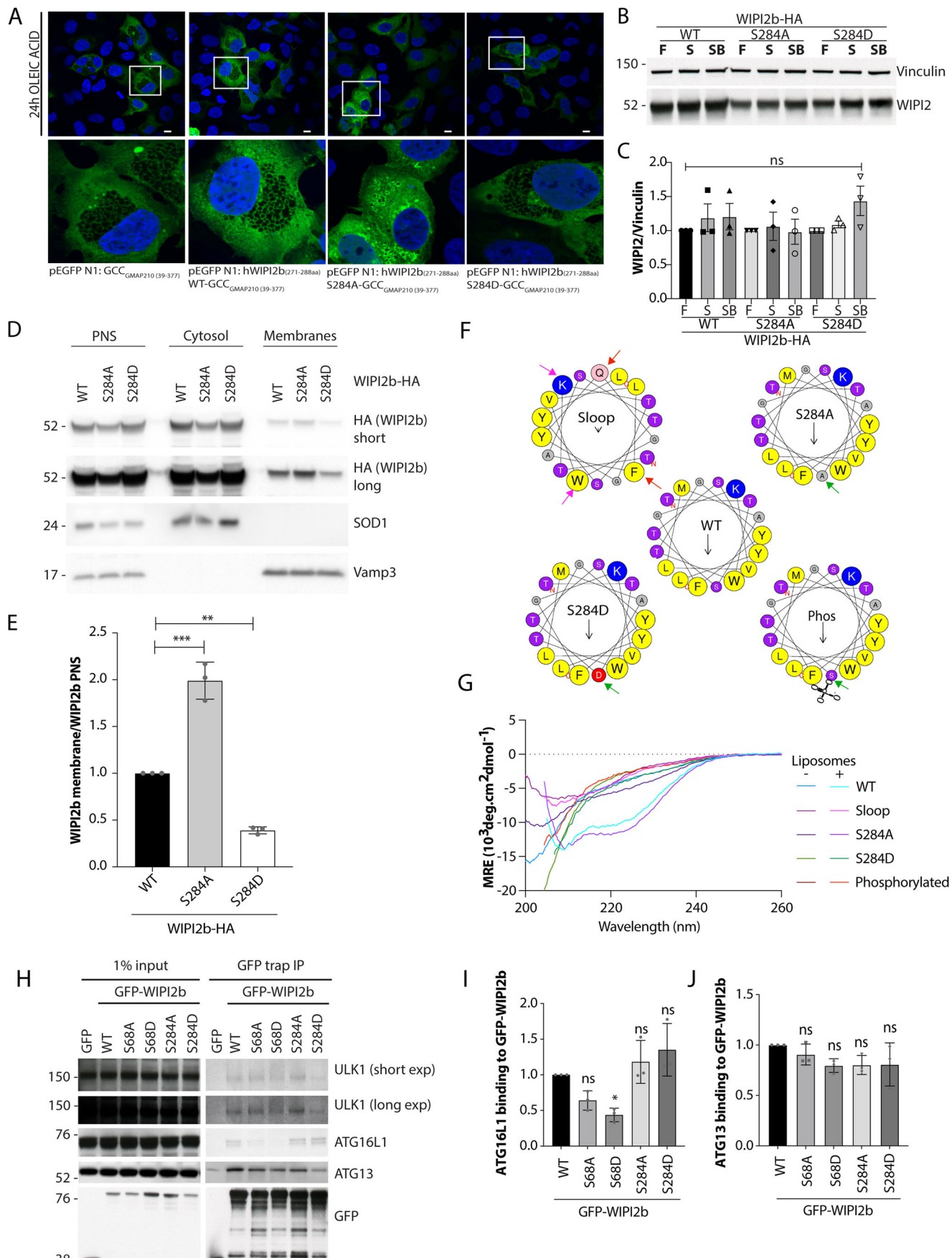

**Figure EV4.  Phosphorylation at WIPI2b S284 affects amphipathic helix formation.**

(A) WIPI2 CRISPR KO cells were transiently transfected cells with pEGFP N1:GCC-GMAP210 (39-377) (negative control), pEGFP N1:WIPI2b WT-GMAP210 (39-377), pEGFP N1:WIPI2b S284A-GMAP210 (39-377), or pEGFP N1:WIPI2b S284D-GMAP210 (30–377) were treated with oleic acid for 24 h and fixed in 4% PFA. Scale bar = 10 μm. (B) WIPI2 KO HEK293A cells transiently expressing WIPI2b-HA WT, WIPI2b-HA S284A or WIPI2b-HA S284D were left untreated (Fed, F) incubated in EBSS for 2 h with (SB) or without Bafilomycin A1 (S). (C) Quantification of WIPI2 levels in (B) normalised to vinculin. SEM from $n = 3$ biological replicates. Statistical analysis was performed by one-way ANOVA with Tukey's post test, ns, $P > 0.05$. (D) Crude cell fractionation was performed in HEK293A cells transiently expressing WIPI2b-HA WT, WIPI2b-HA S284A and WIPI2b-HA S284D. Equal volumes of post-nuclear supernatant (PNS), cytosol and membranes were analysed by western blot. Representative experiment of $n = 3$. (E) Quantification of WIPI2b-HA membrane levels in (D). SEM from $n = 3$ biological replicates. Statistical analysis was performed by one-way ANOVA with Tukey's post test. **$P = 0.0017$, ***$P = 0.001$. (F) Helical wheel representations of WT and amphipathic helix peptides used for CD spectra (made in Heliquest (Gautier et al, 2008)). Coloured arrows indicate where mutations/modifications have been made. Residues are colour coded as follows; blue, basic; red, acidic; yellow, hydrophobic; purple, serine and threonine; pink, asparagine and glutamine; grey, alanine and glycine. (G) Far UV CD spectra of individual peptides in the absence and presence of 3 mM 100 nm liposomes. Raw CD data is converted to mean residue ellipticity and the percentage helicity is calculated from the value at 222 nm. $N = 3$. (H) HEK293A cells transiently expressing GFP alone, GFP-WIPI2b WT, GFP-WIPI2b S68A, GFP-WIPI2b S68D, GFP-WIPI2b S284A or GFP-WIPI2b S284D were lysed and subjected to GFP-TRAP co-immunoprecipitation. Protein complexes were resolved by SDS-PAGE and analysed by western blot. (I) ATG16L1 binding to GFP-WIPI2b quantified using one-way ANOVA with Tukey's post test. SEM from $n = 3$ biological replicates. *$P = 0.0381$. (J) ATG13 binding to GFP-WIPI2b quantified using one-way ANOVA with Tukey's post test. SEM from $n = 3$ biological replicates. ns, $P > 0.05$.

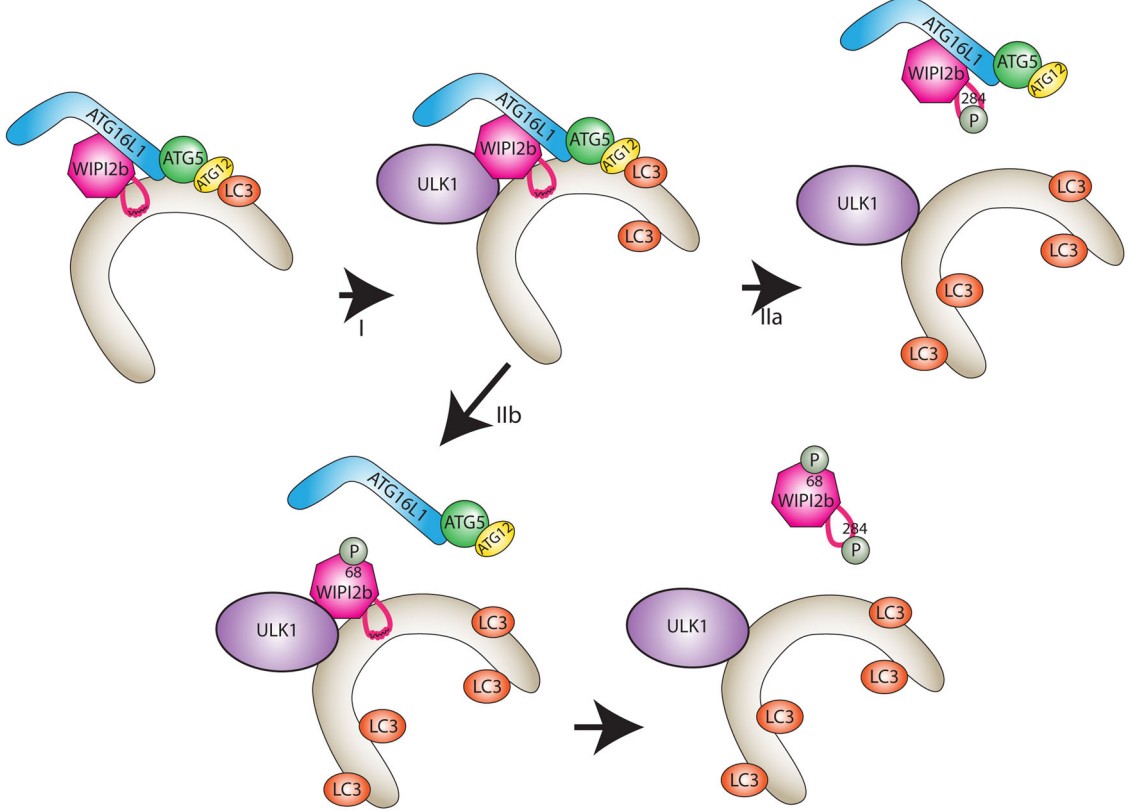

**Figure EV5. Working model.**

Upon autophagy initiation WIPI2b is recruited to PI3P-positive membranes, which in turn recruits ATG16L1 for LC3 lipidation. (I) ULK1 binds WIPI2b and (IIa) phosphorylates it at S284 and (IIb) S68. Phosphorylation at S284 reduces WIPI2bs membrane association, likely by disrupting amphipathic helix formation.

