## [Peer Review File · EMBO Reports]

WIPI2b recruitment to phagophores and ATG16L1 binding are regulated by ULK1 phosphorylation

Andrea Gubas, Eleanor Attridge, Harold B. J. Jefferies, Taki Nishimura, Minoos Razi, Simone Kunzelmann, Yuval Gilad, Thomas J. Mercer, Michael M. Wilson, Adi Kimchi and Sharon A. Tooze

Corresponding author(s): Sharon Tooze (Sharon.tooze@crick.ac.uk)

Review Timeline:

Submission Date:	15th Dec 23
Editorial Decision:	8th Jan 24
Revision Received:	2nd May 24
Editorial Decision:	5th Jun 24
Revision Received:	21st Jun 24
Accepted:	4th Jul 24

Transaction Report:

Dear Sharon,

Happy New Year! We have meanwhile received the full set of referee reports for your manuscript (copied below).

As you will see, all three referees consider the data overall convincing and of interest to a broader audience, but they also raise a number of concerns and ask for further evidence that ULK1 directly phosphorylates WIPI2d in vitro and that the phosphorylation occurs under endogenous conditions.

Given these supportive and constructive comments, we would like to invite you to revise your manuscript with the understanding that the referee concerns (as detailed above and in their reports) must be fully addressed and their suggestions taken on board. Please address all referee concerns in a complete point-by-point response. Acceptance of the manuscript will depend on a positive outcome of a second round of review. It is EMBO Reports policy to allow a single round of revision only and acceptance or rejection of the manuscript will therefore depend on the completeness of your responses included in the next, final version of the manuscript.

We realize that it is difficult to revise to a specific deadline. In the interest of protecting the conceptual advance provided by the work, we recommend a revision within 3 months (April 8th). Please drop me a note ahead of this time if you require more time to complete the revisions.

I am also happy to discuss the revision further via e-mail or a video call, if you wish.

As it stands, your article will be published in our Reports section (studies with up to 5 figures). For short reports, the revised manuscript should not exceed 27,000 characters (including spaces but excluding materials & methods and references) and 5 main plus 5 expanded view figures. The results and discussion sections must further be combined, which will help to shorten the manuscript text by eliminating some redundancy that is inevitable when discussing the same experiments twice. For a normal article there are no length limitations, but it should have more than 5 main figures and the results and discussion sections must be separate. In both cases, the entire materials and methods must be included in the main manuscript file.

*****IMPORTANT NOTE:

We perform an initial quality control of all revised manuscripts before re-review. Your manuscript will FAIL this control and the handling will be delayed IN CASE the following APPLIES:

- 1) A data availability section providing access to data deposited in public databases is missing. If you have not deposited any data, please add a sentence to the data availability section that explains that.
- 2) Your manuscript contains statistics and error bars based on $n=2$. Please use scatter blots in these cases. No statistics should be calculated if $n=2$.

When submitting your revised manuscript, please carefully review the instructions that follow below. Failure to include requested items will delay the evaluation of your revision.*****

- 1) a .docx formatted version of the manuscript text (including legends for main figures, EV figures and tables). Please make sure that the changes are highlighted to be clearly visible.
- 2) individual production quality figure files as .eps, .tif, .jpg (one file per figure). Please download our Figure Preparation Guidelines (figure preparation pdf) from our Author Guidelines pages <https://www.embopress.org/page/journal/14693178/authorguide> for more info on how to prepare your figures.
- 3) a .docx formatted letter INCLUDING the reviewers' reports and your detailed point-by-point responses to their comments. As part of the EMBO Press transparent editorial process, the point-by-point response is part of the Review Process File (RPF), which will be published alongside your paper.
- 4) a complete author checklist, which you can download from our author guidelines (<<https://www.embopress.org/page/journal/14693178/authorguide>>). Please insert information in the checklist that is also reflected in the manuscript. The completed author checklist will also be part of the RPF.
- 5) Please note that all corresponding authors are required to supply an ORCID ID for their name upon submission of a revised manuscript (<<https://orcid.org/>>). Please find instructions on how to link your ORCID ID to your account in our manuscript

tracking system in our Author guidelines

(<<https://www.embopress.org/page/journal/14693178/authorguide#authorshipguidelines>>)

6) We replaced Supplementary Information with Expanded View (EV) Figures and Tables that are collapsible/expandable online. A maximum of 5 EV Figures can be typeset. EV Figures should be cited as 'Figure EV1, Figure EV2' etc... in the text and their respective legends should be included in the main text after the legends of regular figures.

7) Please note that a Data Availability section at the end of Materials and Methods is now mandatory. In case you have no data that requires deposition in a public database, please state so instead of refereeing to the database.

See also < <https://www.embopress.org/page/journal/14693178/authorguide#dataavailability>>. Please note that the Data Availability Section is restricted to new primary data that are part of this study.

Additional information on source data and instruction on how to label the files are available

<<https://www.embopress.org/page/journal/14693178/authorguide#sourcedata>>.

10) Figure legends and data quantification:

- the name of the statistical test used to generate error bars and P values,
- the number (n) of independent experiments (please specify technical or biological replicates) underlying each data point,
- the nature of the bars and error bars (s.d., s.e.m.)

- If the data are obtained from n {less than or equal to} 5, show the individual data points in addition to the SD or SEM.

- If the data are obtained from n {less than or equal to} 2, use scatter blots showing the individual data points.

11) Our journal encourages inclusion of *data citations in the reference list* to directly cite datasets that were re-used and obtained from public databases. Data citations in the article text are distinct from normal bibliographical citations and should directly link to the database records from which the data can be accessed. In the main text, data citations are formatted as follows: "Data ref: Smith et al, 2001" or "Data ref: NCBI Sequence Read Archive PRJNA342805, 2017". In the Reference list, data citations must be labeled with "[DATASET]". A data reference must provide the database name, accession number/identifiers and a resolvable link to the landing page from which the data can be accessed at the end of the reference. Further instructions are available at <<https://www.embopress.org/page/journal/14693178/authorguide#referencesformat>>.

12) All Materials and Methods need to be described in the main text. We would encourage you to use 'Structured Methods', our new Materials and Methods format. According to this format, the Materials and Methods section should include a Reagents and Tools Table (listing key reagents, experimental models, software and relevant equipment and including their sources and relevant identifiers) followed by a Methods and Protocols section in which we encourage the authors to describe their methods using a step-by-step protocol format with bullet points, to facilitate the adoption of the methodologies across labs.

More information on how to adhere to this format as well as downloadable templates (.doc or .xls) for the Reagents and Tools Table can be found in our author guidelines: <<https://www.embopress.org/page/journal/14693178/authorguide#manuscriptpreparation>>. An example of a Method paper with Structured Methods can be found here: <<https://www.embopress.org/doi/10.15252/msb.20178071>>.

13) As part of the EMBO publication's Transparent Editorial Process, EMBO Reports publishes online a Review Process File to accompany accepted manuscripts. This File will be published in conjunction with your paper and will include the referee reports, your point-by-point response and all pertinent correspondence relating to the manuscript.

Best wishes,

Martina

Referee #1:

In this interesting study, Gubas A et al. demonstrated that ULK1 regulates the recruitment of WIPI2b to phagophores and the binding of ATG16L1 to negatively regulate autophagy, employing tools from cell biology, biochemistry, and biophysics. The results are convincing, and the claimed points are clearly illustrated. However, several issues need to be addressed before publication.

Major:

1. In Fig 4B and Fig 5A, ULK1 phosphorylates WIPI2 at S68 and S284 in vivo, respectively. The direct phosphorylation of full-length WIPI2 by ULK1 at S68 and S284 needs validation in in vitro phosphorylation assays, either by observing a band shift or by examining with an antibody against pS284.
2. The phosphorylation of WIPI2 at S68 and S284 inhibits starvation-induced autophagy. Since WIPI2 also functions in other types of autophagy, such as mitophagy and STING activation-induced non-canonical autophagy, it is important to determine whether this regulation is also present in these two types of autophagy.
3. As mentioned in the manuscript, K88E and K128E mutants failed to be recruited to PI3P-enriched liposomes. Is it possible for the S68 mutant? This can also be tested using CD spectra.
4. Does ULK2 have a similar function in the phosphorylation of WIPI2? At the very least, this should be discussed.

Minor:

1. The description of LC3/GABARAP proteins is not accurate for ATG8 family proteins.
2. Regarding the sentence "Using the structure of WIPI2d[31], we determined that S68 and S284 were good candidates, respectively, for regulating these specific activities," it is better to show the principle or the possible reasons with a cartoon of WIPI2d structure.
3. "To further support direct phosphorylation of WIPI2 by ULK1, we treated cells with SBI-0206965 [36] and MRT68921 [37], two commercially available small molecule inhibitors of ULK1." In vivo assays with these two inhibitors cannot support the direct phosphorylation of WIPI2 by ULK1.
4. "Phosphorylated proteins have a greater negative charge and will hence migrate slower on SDS-PAGE gel than non-phosphorylated proteins." This is not always the case for phosphorylated proteins.
5. The phosphorylation of the other two autophagy-related proteins, ATG9A and STX17, can be included in the Introduction part to show the complete view of ULK in autophagy.

Referee #2:

In the manuscript by Gubas et al, the authors examine the role of ULK1 in phosphorylating WIPI2b. Using an overexpression system, the authors identify multiple candidate ULK1 phosphorylation sites and characterise two of these. Phosphorylation of WIPI2b at S68 disrupts WIPI-ATG16L1 interaction, while phosphorylation at S284 negatively affects WIPI2b membrane association. The second site is characterised in more depth, including generation of a phospho-specific antibody and the molecular analyses in Fig.5 are particularly elegant. The authors build up a model whereby ULK1 potentially coordinates disassembly of the autophagy initiation machinery, via WIPI2b phosphorylation. Overall, this is a solid set of data that adds mechanistic insight into the role of ULK1 in autophagy initiation. However, my main concerns are that all the data is generated with overexpressed ULK1, which could lead to artefactual phosphorylation, especially as ULK1 regulatory complex members are likely not in the correct binding stoichiometry.

Main Points:

- 1) Can any of these WIPI2b phosphosites be identified under endogenous conditions (i.e. no overexpression of ULK1)? Perhaps the authors could consider some of the following: A) Can mass spec of endogenous WIPI IP identify them in WT vs ULK1/2 knockouts (or ATG13/FIP200 KOs)? B) I note that the authors have a phospho-specific antibody to S284 - but this is only tested in their overexpression system. Does it work with endogenous protein (the authors may need to IP WIPI2 or even use the phospho-specific antibody to IP)? C) In ULK1/2 KO cells is there an observable shift in mobility of endogenous WIPI2b (maybe a phos-tag gel might help here)?
- 2) I was a little confused with the S395 phosphorylation data, as it appears mutation of this residue impairs ULK1-mediated WIPI phosphorylation (less of a shift in WIPI2 mobility (Fig.S2))? Why do the authors think this is the case, especially as this site should be lost upon autophagy induction?
- 3) Based on authors data, overexpression of ULK1 should increase WIPI phosphorylation. Therefore, in Fig.4E and F, does overexpression of WT ULK alter ATG16L1 interaction/puncta and is this dependent on S68 phosphorylation (i.e. not seen in S68A)?
- 4) Similar to above, in Fig.5 B and C, does ULK1 overexpression block WT WIPI2 puncta but not S284A WIPI2?
- 5) Also in Fig.5C, why is BafA1 increasing WIPI2 puncta in the S284A mutant? The authors do briefly mention in the discussion that this could be due to retention on autophagosomes and lysosome delivery. This should be straightforward to test by western blot and could provide strong evidence to support their model.

Referee #3:

Autophagosome biogenesis is mediated by a conserved set of protein modules, collectively referred to as the autophagy machinery. While progress has been made in the recent years with regard to how the autophagy machinery acts to mediate the formation of autophagosomes, major questions concerning their mechanisms of action, dynamics and regulation remain. Here, the authors investigate the regulation of WIPI2 by the ULK1 kinase, two components of the autophagy machinery. Combining cell biological, biochemical and biophysical approaches, the authors show that phosphorylation of WIPI2 by ULK1 negatively regulates the association of WIPI2 with nascent autophagosomal membranes. In particular, phosphorylation of S68 reduces binding to ATG16L1, whereas the phosphorylation of S284 inhibits membranes binding. The latter effect is likely due to the inhibition of the formation of a membrane inserting amphipathic helix.

Overall, this is a well conducted, convincing study that sheds light on the regulation of the autophagy machinery during autophagosome biogenesis. It will therefore be of interest to the wider community studying this process.

Below, I have a few comments the authors should address before publication.

1. The authors claim that the interaction of the kinase dead ULK1 (ULK1 KI, Figure 1F) with WIPI2 is reduced in comparison to the wt protein. However, the expression level of this mutant is also lower. Similar concerns apply to Figures 2D,G and S2C,D. It is not unusual that kinase dead versions of proteins kinase have lower expression levels and this effect may therefore not be avoidable by the authors. Nevertheless, the corresponding conclusions should be tone down.
2. The inhibitory effect of WIPI2 KO on autophagic flux as measured by LC3B-II and p62 stabilization is relatively small. For example, is there any significant difference between the EV condition for the WIPI2 KO compared to the EV condition for the wt cells as well as the rescue with the wt WIPI2 construct in terms of LC3B-II/vinculin levels for Figure 3A/B? The authors should elaborate on the potential reasons for the relatively minor effect of WIPI2 depletion on autophagic flux.
3. Figure S3 A,B and page 9 bottom: The authors write that there is a trend towards stronger membrane binding by the WIPI2b 6A mutant and a weaker membrane association for the 6D mutant. The authors should consider repeating this experiment in order to see if this will allow the authors to make a stronger statement. Also, I understand that the 6A/6D mutant contains the mutation of S284. When this site is mutated in isolation a very strong effect is seen on membrane binding (Figure 5D,E). How can this discrepancy be explained?
4. Figure 3E, page 10: The authors extensively elaborate on the results of the phosphosite peptide array screen but the actual take home message is not really clear. It is advisable to shorten or rephrase the respective paragraph. In its present

representation it is not obvious what these data and the discussion add to the manuscript.

Minor: There are two brackets misplaced in the middle of page 11.

We thank all the referees for their positive and helpful comments. We have addressed all the comments as detailed below. The additional data and corrections have improved the manuscript significantly. Our responses are in italics.

Referee #1:

In this interesting study, Gubas A et al. demonstrated that ULK1 regulates the recruitment of WIPI2b to phagophores and the binding of ATG16L1 to negatively regulate autophagy, employing tools from cell biology, biochemistry, and biophysics. The results are convincing, and the claimed points are clearly illustrated. However, several issues need to be addressed before publication.

Major:

1. In Fig 4B and Fig 5A, ULK1 phosphorylates WIPI2 at S68 and S284 **in vivo**, respectively. The direct phosphorylation of full-length WIPI2 by ULK1 at S68 and S284 needs **validation in in vitro phosphorylation assays**, either by observing a band shift or by examining with an antibody against pS284.

We appreciate the comment and the requirement for demonstration of a direct phosphorylation of WIPI2b by ULK1. We have performed an in vitro kinase assay with myc-ULK1 (WT or KI) immunoprecipitated from cells and purified WIPI2b. MS site identification confirmed S68 and S284 on purified WIPI2b with the WT kinase (see new Figure 5C and D, and text on page 8).

2. The phosphorylation of WIPI2 at S68 and S284 inhibits starvation-induced autophagy. Since WIPI2 also functions in other types of autophagy, such as mitophagy and STING activation-induced non-canonical autophagy, it is important to determine whether this regulation is also present in these two types of autophagy.

We agree with the Referee that this is a very important question. We carefully considered the questions concerning both NIX and STING, and in particular the model systems and data required to properly address these questions. We feel that addressing this is beyond the scope of this EMBO Reports manuscript. We omitted to mention the publication showing the regulation of STING by WIPI2 and have added the reference, Wan et al., 2023 and speculate that S284 may disrupt binding of STING and WIPI2 (text edited on page 18).

3. As mentioned in the manuscript, K88E and K128E mutants failed to be recruited to PI3P-enriched liposomes. Is it possible for the S68 mutant? This can also be tested using CD spectra.

We have added text to address this point (page 13). Strong et al 2021b (Figure 6) had observed that K88E and K128 had reduced binding in vitro to PI3P-enriched liposomes. We speculate that the observation we made with the S68D mutant having reduced ATG16L1 puncta in cells (Figure 4F) may be related to the observations of Strong et al. in vitro. However, as Strong concluded (and we agree) it is not clear why these mutants (which are not near the membrane associated regions) disrupt membrane association. We could not design an experiment using CD to test this point as we surmised that the region of WIPI2b which contains S68 may not adopt a secondary structure suitable for CD analysis.

However, to explore this further for the Referee we performed a membrane fractionation experiment with WIPI2b-S68A/D in parallel with 6A and 6D with 284D. This data (Figure 1R for referee, n=3) confirms the lack of 284D membrane association. Data with mutations of 6A, 6D, (see Figure EV2F) and here attached Figure 1R for referee only show a slight increase for 6A (but non-significant when all mutants are considered in statistical analysis) in membrane association compared to WT. 68A and 68D do not result in any significant change in membrane association but again the most striking is the significant reduction in S284D. Thus, the observation Strong et al have made about the effect of K88E and K128E on PI3P binding, and the lack of puncta in the cells rescued with S68D is not reflected by altered membrane association in cells and likely has to do with the recruitment of ATG16L1 influencing (stabilizing?) WIPI2 at PI3P-positive structures.

Referee Figure 1R. Crude membrane fractionation of WIPI2KO cells overexpressing WIPI2b-HA WT, 6A, 6D, 68A, 68D or 284D. Quantification of mean with SEM. One-way ANOVA with Dunnetts multiple comparison test * $p < 0.05$ n=3.

4. Does ULK2 have a similar function in the phosphorylation of WIPI2? At the very least, this should be discussed.

We have discussed the role of ULK2 on Page 4, and cited Demeter et al 2020 which suggests they have distinct activities.

We also carried out a number of experiments in the ULK1/2 DKO MEFs cell line we originally isolated. In the absence of ULK2 antibodies we do not have a valid way of generating ULK2-specific knock out in cell lines. We choose to initially analyse the phosphorylation status in WT and DKO MEFs using phostag gels because of the difficulty in isolating the effect of the phosphorylation sites on WIPI2b with our existing reagents which are pan-WIPI2 or the phosphospecific S284 which does not detect endogenous WIPI2 S284 phosphorylation by western blot. The data is attached below for the Referee Figure 2R. There was a potential change in the band pattern in N=1 and more clearly in N=3 (marked by red arrow) in DKO compared to WT but unfortunately the results were not robust or reproducible comparing WT and DKO with this approach, so we did not test the WT MEFs compared to the ULK1 single KO MEFs.

Referee Figure 2R. MEF WT or ULK1/2 DKO were incubated in full medium (F) for 1 hour or starved (S) for 30 mins and 1 hour. In repetition N=4 cells were also treated with MRT68921. Cells were lysed in TNTE (1% triton, 20mM Tris/HCl pH 7.5, 5mM EDTA, 150mM NaCl supplemented with 1x Complete protease inhibitor cocktail (Roche) and 1x PhosSTOP (Roche)) After 5x Laemmli sample buffer was added samples were supplemented with 10mM MnCl₂. 8% Bis-Tris gels were made with 50mM phostag additive (ApexBio) and 100mM MnCl₂. Gel mix was de-gassed for 10 minutes before pouring. Gels were run at 70V until stacked and then 130V for a total time of 2.5hrs. Gels were washed 3X 20-minutes with transfer buffer supplemented with 10mM EDTA and 0.05% SDS and then 1x20-minutes with 0.05% SDS transfer buffer before transfer. Gels were subjected to immunoblotting staining with anti-WIPI2 (monoclonal Ms), anti-WIPI2 (polyclonal Rb) or anti-ULK1 (Rb). Figure shows 3 independent experiments with 2 technical repeats per biological replicate, N=3 samples were run twice on separate gels. Note lysis buffer with EGTA instead of EDTA or without EDTA/EGTA was tested (N=2) but band shifts were not observed (data not shown). Red mark on top gel in N=3 indicates the possible difference in WT versus DKO MEFS.

Minor:

1. The description of LC3/GABARAP proteins is not accurate for ATG8 family proteins. The text has been edited: The abstract, page 4 and 17 to accurately describe the LC3 and GABARAP proteins as belonging to the ATG8 family.

2. Regarding the sentence "Using the structure of WIPI2d[31], we determined that S68 and S284 were good candidates, respectively, for regulating these specific activities," it is better to show the principle or the possible reasons with a cartoon of WIPI2d structure.

We agree the previous version was confusing and we have removed the last sentence (originally 1st paragraph, last sentence, page 11) and rephrased the rationale for choosing the two phosphorylation sites (see page 11, second paragraph). In Figure EV3A we have modelled the phosphate group on S68 in WIPI2d and shown the predicted change in interaction with ATG16L1 α -helix.

3. "To further support direct phosphorylation of WIPI2 by ULK1, we treated cells with SBI-0206965 [36] and MRT68921 [37], two commercially available small molecule inhibitors of ULK1." In vivo assays with these two inhibitors cannot support the direct phosphorylation of WIPI2 by ULK1.

We have changed the text to remove "direct" on page 8.

4. "Phosphorylated proteins have a greater negative charge and will hence migrate slower on SDS-PAGE gel than non-phosphorylated proteins." This is not always the case for phosphorylated proteins.

*We have changed the text on page 7. It now reads "Phosphorylated proteins have a greater negative charge **and can migrate** slower on SDS-PAGE gel than non-phosphorylated proteins."*

5. The phosphorylation of the other two autophagy-related proteins, ATG9A and STX17, can be included in the Introduction part to show the complete view of ULK in autophagy.

We have added these details to the Introduction on page 3 and 4, along with appropriate references Zhou et al. [10.1038/cr.2016.146](https://doi.org/10.1038/cr.2016.146) and Wang et al., [10.1083/jcb.202211025](https://doi.org/10.1083/jcb.202211025).

Referee #2:

In the manuscript by Gubas et al, the authors examine the role of ULK1 in phosphorylating WIPI2b. Using an overexpression system, the authors identify multiple candidate ULK1 phosphorylation sites and characterise two of these. Phosphorylation of WIPI2b at S68 disrupts WIPI-ATG16L1 interaction, while phosphorylation at S284 negatively affects WIPI2b membrane association. The second site is characterised in more depth, including generation of a phospho-specific antibody and the molecular analyses in Fig.5 are particularly elegant. The authors build up a model whereby ULK1 potentially coordinates disassembly of the autophagy initiation machinery, via WIPI2b phosphorylation. Overall, this is a solid set of data that adds mechanistic insight into the role of ULK1 in autophagy initiation. However, my main concerns are that all the data is generated with overexpressed ULK1, which could lead to artefactual phosphorylation, especially as ULK1 regulatory complex members are likely not in the correct binding stoichiometry.

Main Points:

1) Can any of these WIPI2b phosphosites be identified under endogenous conditions (i.e. no overexpression of ULK1)? Perhaps the authors could consider some of the following: A) Can mass spec of endogenous WIPI IP identify them in WT vs ULK1/2 knockouts (or ATG13/FIP200 KOs)? B) I note that the authors have a phospho-specific antibody to S284 - but this is only tested in their overexpression system. Does it work with endogenous protein (the authors may need to IP WIPI2 or even use the phospho-specific antibody to IP)? C) In ULK1/2 KO cells is there an observable shift in mobility of endogenous WIPI2b (maybe a phos-tag gel might help here)?

We agree this is a major point, and of fundamental importance. However, detection of endogenous phosphorylation is challenging. We thank the referees for the suggestions. This was very helpful.

We performed all three suggestions with varying success. We were successful in being able to immunoprecipitate phosphorylated S284 with our phosphospecific antibody and have included this data in Figure 5, panel B. To do this we had to treat the cells with Okadaic acid to inhibit phosphatases. We were also able to prove the phosphorylation of WIP2b at S284 by ULK1 was direct in new Figure 5C and D, and we used this experimental set up to re-validate S68 and S284 using mass spectroscopy (see text page 8).

We also tried phos-tag gels (see Figure 2R above) which were not successful.

2) I was a little confused with the S395 phosphorylation data, as it appears mutation of this residue impairs ULK1-mediated WIPI phosphorylation (less of a shift in WIPI2 mobility (Fig.S2))? Why do the authors think this is the case, especially as this site should be lost upon autophagy induction?

On Page 9 we have rewritten the text to better explain the S395 data and removed the data about ULK1 phosphorylation of the S395 mutant (see edited Figure EV 2E). We hope this is helpful. We believe there might be a reciprocal relationship ULK1 phosphorylation and mTORC1 phosphorylation but have not been able to obtain a clear mechanism. We attach the figure below for the referee's information showing the increase in mobility of the S395-positive WIPI2 band in the 6D mutants.

Referee Figure 3R. HEK293A cells co-expressing WIPI2b-HA WT, WIPI2b-HA S6D or WIPI2b-HA S395A, and myc-ULK1 (where indicated) were incubated for 2 hours in EBSS. Cells were lysed and lysate was treated where indicated with lambda phosphatase and analysed by western blot.

3) Based on authors data, overexpression of ULK1 should increase WIPI phosphorylation. Therefore, in Fig.4E and F, does over expression of WT ULK alter ATG16L1 interaction/puncta and is this dependent on S68 phosphorylation (i.e. not seen in S68A)?

We thank the author for this question. We have not done any experiments looking at puncta, or flux, or ATG16L1 interaction with overexpressed ULK1. We have published in Chan et al, JBC 2007 that overexpression of ULK1 can itself inhibit LC3 lipidation, and long-lived protein degradation so we do not think this experiment would be easy to interpret.

4) Similar to above, in Fig.5 B and C, does ULK1 overexpression block WT WIPI2 puncta but not S284A WIPI2?

Please see above comment.

5) Also in Fig.5C, why is BafA1 increasing WIPI2 puncta in the S284A mutant? The authors do briefly mention in the discussion that this could be due to retention on autophagosomes and lysosome delivery. This should be straightforward to test by western blot and could provide strong evidence to support their model.

We thank the author for this suggestion. As suggested, we analysed the levels of WIPI2 in fed, starved and starved with BafA1 (see Figure 4R for referee). We did not observe any changes in the levels of WIPI2. We have edited the text in the Discussion on page 19 to reflect this data and our current hypothesis.

Referee Figure 4R. *WIPI2 CRISPR KO cells stably expressing WIPI2 WT, S284A or S284D were incubated for 2 hours in fed, EBSS (S) or EBSS with Bafilomycin A. Cells were harvested, lysed and subjected to SDS-PAGE and western blotted with anti-WIPI2 antibody.*

Referee #3:

Autophagosome biogenesis is mediated by a conserved set of protein modules, collectively referred to as the autophagy machinery. While progress has been made in the recent years with regard to how the autophagy machinery acts to mediate the formation of autophagosomes, major questions concerning their mechanisms of action, dynamics and regulation remain. Here, the authors investigate the regulation of WIPI2 by the ULK1 kinase, two components of the autophagy machinery. Combining cell biological, biochemical and biophysical approaches, the authors show that phosphorylation of WIPI2 by ULK1 negatively regulates the association of WIPI2 with nascent autophagosomal membranes. In particular, phosphorylation of S68 reduces binding to ATG16L1, whereas the phosphorylation of S284 inhibits membranes binding. The latter effect is likely due to the inhibition of the formation of a membrane inserting amphipathic helix.

Overall, this is a well conducted, convincing study that sheds light on the regulation of the autophagy machinery during autophagosome biogenesis. It will therefore be of interest to the wider community studying this process.

Below, I have a few comments the authors should address before publication.

1. The authors claim that the interaction of the kinase dead ULK1 (ULK1 KI, Figure 1F) with WIPI2 is reduced in comparison to the wt protein. However, the expression level of this mutant is also lower. Similar concerns apply to Figures 2D,G and S2C,D. It is not unusual that kinase dead versions of proteins kinase have lower expression levels and this effect may therefore not be avoidable by the authors. Nevertheless, the corresponding conclusions should be tone down.

We agree with this point and despite numerous titrations the KI often is expressed less. We have minimized the conclusion about the lack of effect of the KI (see page 8 and added to the Figure legend 1 that the quantification of the interaction of the ULK1 with WIPI2 was normalized (page 31).

2. The inhibitory effect of WIPI2 KO on autophagic flux as measured by LC3B-II and p62 stabilization is relatively small. For example, is there any significant difference between the EV condition for the WIPI2 KO compared to the EV condition for the wt cells as well as the rescue with the wt WIPI2 construct in terms of LC3B-II/vinculin levels for Figure 3A/B? The authors should elaborate on the potential reasons for the relatively minor effect of WIPI2 depletion on autophagic flux.

We agree with the comments by the referee. In the model system of WIPI2 KO, even with CRISPR, in it always difficult to show a robust inhibition of LC3 lipidation and p62 degradation. This is well documented in publications from several labs. We have selected the best clones with the maximum modulatory phenotype (see Figure EV1) which have reduced LC3 lipidation and inhibition of p62 degradation. The major issue with knock out of WIPI2 is that there are 3 other WIPI proteins in cells: WIPI1, 3 and 4. WIPI1 has been shown to be required for autophagy (Mauthe et al., Autophagy 2011) and the presence of WIPI1 likely lessens the effect of loss of WIPI2. There may be pathways that feed into autophagy that are supported by WIPI1, 3 and 4 proteins which also compensate for the loss of WIPI2.

3. Figure S3 A,B and page 9 bottom: The authors write that there is a trend towards stronger membrane binding by the WIPI2b 6A mutant and a weaker membrane association for the 6D

mutant. The authors should consider repeating this experiment in order to see if this will allow the authors to make a stronger statement. Also, I understand that the 6A/6D mutant contains the mutation of S284. When this site is mutated in isolation a very strong effect is seen on membrane binding (Figure 5D,E). How can this discrepancy be explained?

We thank the referee for this suggestion. We have repeated this experiment, which is now Figure S3A, B is Figure EV2F and G. We have replaced the old data with this new set of data. There is now a statistically significant difference between WT and 6A mutant. However, we feel it is important to note that WIPI2 puncta is a more reliable readout for WIPI2 function and membrane association. For example, new Figure EV2F and G confirms 6D is associated with membranes where in contrast in Figure 3E there are very few puncta. The interpretation just based on membrane association would be that there is no effect, but we are confident that puncta formation represents the true phenotype. Note, all the membrane fractionation experiments with the phosphomutants have been done in fed medium as we previously showed that there is no difference in total membrane association between fed and starved, revealing membrane association does not correlate to puncta formation (Polson et al., Autophagy 6:4, 506-522; 2010).

Regarding the last point, we have edited the text to explain our thoughts on the 6A and 6D constructs. See page 11 where we note the data with the 6 mutations could add a lot of negative charge, and the structure might change slightly, which may affect membrane association.

4. Figure 3E, page 10: The authors extensively elaborate on the results of the phosphosite peptide array screen but the actual take home message is not really clear. It is advisable to shorten or rephrase the respective paragraph. In its present representation it is not obvious what these data and the discussion add to the manuscript.

Given the results of the phosphopeptide analysis, which just supported S284 phosphorylation, were not very relevant to the manuscript once we demonstrated that the endogenous S284 was phosphorylated under starvation we decided to remove the phosphopeptide peptide array.

Minor: There are two brackets misplaced in the middle of page 11. **Done, now page 12.**

Dear Sharon,

Thank you for the submission of your revised manuscript to EMBO Reports. We have now received the full set of referee reports that is copied below.

As you will see, all referees are very positive about the study and support publication. Before I can formally accept your manuscript, I kindly ask you to address a few editorial things, as listed below:

- Your manuscript will be published in our Reports section, which requires a shorter format with a combined Results and Discussion section. The character count should not exceed 27,000 characters (including spaces but excluding materials & methods and references). A few characters more or less does not matter that much, though.
 - Please provide up to 5 keywords.
 - The conflict of interest declaration needs to be moved from the Acknowledgements to a separate section called 'Disclosure and competing interests statement'. For more information see <https://www.embopress.org/page/journal/14693178/authorguide#conflictsofinterest>
 - Please provide the author list in the manuscript list as First Name Last Name, i.e., the other way round as it is now and with the First Name spelled out.
 - Please note that all conclusions need to be substantiated by the underlying data. Therefore, please show/include the mass spectrometry data mentioned on page 8 and the Western blot analysis of WIPI2 levels mentioned on page 19.
 - The funding information provided in the manuscript and in the online manuscript tracking system must match. In this respect we note that the 6 grant numbers provided in the manuscript and in the system do not match, e.g. manuscript file: Cancer Research UK (CC2134 and CC1063) vs. system: Cancer Research UK (FC001187 and FC001999), etc.
 - The manuscript sections should be in the following order: Title page - Abstract & Keywords - Introduction - Results - Discussion - Methods - Data Availability - Acknowledgments - Disclosure Statement & Competing Interests - References - Figure Legends - Tables with legends - Expanded View Figure Legends.
 - Materials and Methods should be Methods
 - ORCID numbers should be removed from the title page
 - The titles of the EV figures in the legends and figure files need to be corrected, e.g. it should be Figure EV1 instead of Expanded view Figure 1
 - Our e-mails to two co-authors bounced: Michael Wilson - michael.wilson@bbsrc.ac.uk and Mino Razi - minoo.razi@crick.ac.uk. Please check and provide their current e-mail addresses, in case these changed.
 - We perform a routine image analysis on all manuscripts before publication. In this context we noticed that the same LC3B Western blot might have been used for Fig 3A and Fig EV1B. Could you please check the composition of these two figure panels? In case the same blot was used intentionally, and all the Western blots originate from one experiment, i.e., the same controls etc. are valid for all Western blots, please note the duplicate use in the figure legend to avoid any ambiguity.
 - The LC3 immunostainings used in Fig. EV1 D and EV3 B appear also similar (control, clone 1). Please check the composition of these figure panels and verify that all images shown correspond to the same experiment. If so, please clearly state in the figure legend that the images shown in these panels are the same and correspond to the same experiment to avoid any ambiguities.
 - Our production/data editors have asked you to clarify several points in the figure legends (see below). Please incorporate these changes in the manuscript and return the revised file with tracked changes with your final manuscript submission.
- A) Statistical test information. Only p-values that are actually shown in the figure panel(s) should (and must) be defined in the legends, all others should be removed from (or added to) the legend. Moreover, we require that the exact p-values are given instead of a range:
- Please note that the exact p values are not provided in the legends of figures 1c, e, g; 3b, d; 4d, f-g; 5e, g, i; EV 1b-c; EV 2g; EV 3d; EV 4c, g.
 - Please note that in figures 1e; 4d, i; 5e; EV 3d; EV 4c; there is a mismatch between the annotated p values in the figure legend

and the annotated p values in the figure file that should be corrected.

B) Replicates and error bars:

- Although 'n' is provided, please describe the nature of entity for 'n' in the legends of figures 1c, e, g; 3b; 4d, i; 5i; EV 4c, g-h.
- Please note that scale bar and its definition are missing for figures EV 3b-c.

- Figure 3D: as far as I understand you counted WIPI2-positive puncta from 10 frames per experiment. Even though 10 frames were counted, I assume that these are derived from one experiment, in which case $n = 1$? If so, statistical analysis should not be performed. Also in 4G and 5G, if 100 cells were counted from one experiment, this would in essence also constitute $n = 1$, or where these measurements derived from several independent experiments, in which case one might have to compare the mean of the means between each condition?

- Finally, EMBO Reports papers are accompanied online by

A) a short (1-2 sentences) summary of the findings and their significance,

B) 2-3 bullet points highlighting key results and

C) a schematic summary figure that provides a sketch of the major findings (not a data image).

Please provide the summary figure as a separate file in PNG or JPG format at a size of 550x300-600 pixels (width x height).

Please note that the size is rather small and that text needs to be readable at the final size. Please send us this information along with the revised manuscript.

- On a different note, I would like to alert you that EMBO Press offers a new format for a video-synopsis of work published with us, which essentially is a short, author-generated film explaining the core findings in hand drawings, and, as we believe, can be very useful to increase visibility of the work. This has proven to offer a nice opportunity for exposure i.p. for the first author(s) of the study. Please see the following link for representative examples and their integration into the article web page:

<https://www.embopress.org/doi/full/10.15252/emj.2019103932>

With kind regards,

Martina

Martina Rembold, PhD

Senior Editor

EMBO reports

Referee #1:

This manuscript represents a high-quality study that offers valuable insights in the field of autophagy. During the revision process, the majority of my inquiries have been satisfactorily addressed. It's understandable that a few questions remain unresolved due to technical limitations. I fully support its publication in EMBO Reports.

Referee #2:

The authors have satisfactorily addressed all my concerns.

Referee #3:

The authors have addressed all my comments and I have no further points.

All editorial and formatting issues were resolved by the authors.

Dr. Sharon Tooze
Francis Crick Institute
Molecular Cell Biology of Autophagy
1 Midland Road
London NW1 1AT
United Kingdom

Dear Sharon,

Thank you for submitting your revised manuscript, which I have checked and all looks fine. I am very pleased to accept your manuscript for publication in the next available issue of EMBO reports. Thank you for your contribution to our journal.

Best regards,

Martina
